# MVU-Eval: Towards Multi-Video Understanding Evaluation for Multimodal LLMs

**Tianhao Peng**[*,1,4], **Haochen Wang**[*2], **Yuanxing Zhang**[*3], **Zekun Wang**[3], **Zili Wang**[4],
**Ge Zhang**[4], **Jian Yang**[4], **Shihao Li**[1], **Yanghai Wang**[1], **Xintao Wang**[3], **Houyi Li**[4],
**Wei Ji**[1], **Pengfei Wan**[3], **Wenhao Huang**[4], **Zhaoxiang Zhang**[1,2], **Jiaheng Liu**[†,1]
[1]Nanjing University, [2]CASIA, [3]Kuaishou Technology, [4]M-A-P

## Abstract

The advent of Multimodal Large Language Models (MLLMs) has expanded AI capabilities to visual modalities, yet existing evaluation benchmarks remain limited to single-video understanding, overlooking the critical need for multi-video understanding in real-world scenarios (e.g., sports analytics and autonomous driving). To address this significant gap, we introduce **MVU-Eval**, the first comprehensive benchmark for evaluating **M**ulti-**V**ideo **U**nderstanding for MLLMs. Specifically, our MVU-Eval mainly assesses eight core competencies through 1,824 meticulously curated question-answer pairs spanning 4,959 videos from diverse domains, addressing both fundamental perception tasks and high-order reasoning tasks. These capabilities are rigorously aligned with real-world applications such as multi-sensor synthesis in autonomous systems and cross-angle sports analytics. Through extensive evaluation of state-of-the-art open-source and closed-source models, we reveal significant performance discrepancies and limitations in current MLLMs' ability to perform understanding across multiple videos. The benchmark will be made publicly available to foster future research.

## 1   Introduction

The rise of Large Language Models (LLMs) has enabled numerous groundbreaking applications across various domains. For instance, conversational agents like ChatGPT have revolutionized how we interact with technology by providing coherent and contextually relevant responses in natural language [32]. These models have also shown significant improvements in tasks such as knowledge-based question-answering [14, 36, 10, 3], mathematics [50, 7, 55], and code generation [2, 22, 28].

Multimodal Large Language Models (MLLMs) extend this capability to visual modalities and MLLMs are trained to integrate visual inputs to understand and interpret images and videos [59, 31, 1]. Recently, to evaluate the capability of existing MLLMs for video understanding, many benchmarks have been proposed [49, 21, 12]. However, most of the video understanding benchmarks primarily take a single video as input, which neglects the crucial need for multi-video understanding. *This limitation becomes particularly evident in complex real-world scenarios such as summarization on multiple retrieved relevant videos, sports analytics using various camera angles, or autonomous driving requiring information from multiple cameras.*

To approximate real-world scenarios more accurately, as shown in Figure 1, we introduce the first **M**ulti-**V**ideo Understanding benchmark called **MVU-Eval** [1], which comprehensively assesses **eight** core perception and reasoning abilities through 1,824 carefully curated QA pairs spanning 4,959

---

 * Equal contribution. † Corresponding Author.
 Tianhao Peng was an intern at Nanjing University during the preparation of this manuscript.
[1]https://huggingface.co/datasets/MVU-Eval-Team/MVU-Eval-Data

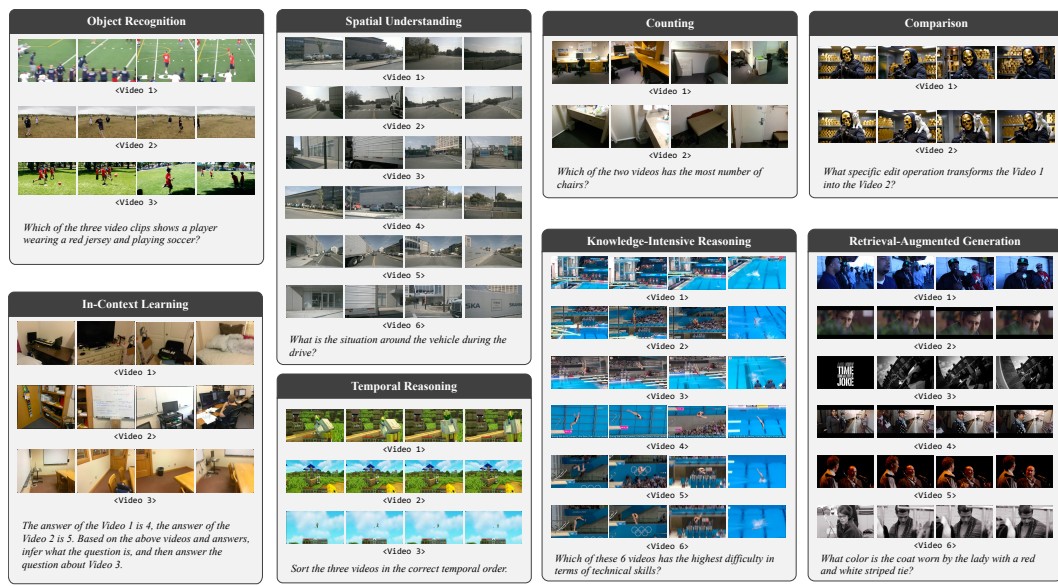

Figure 1: Illustration of several representative examples in our MVU-Eval.

distinct videos from various domains (e.g., life, movie, gaming, autonomous driving). Specifically, the fundamental multi-video perception tasks include Object Recognition (OR), Spatial Understanding (SU), Counting, and Comparison, which aim to evaluate the model's ability to accurately extract the vision feature and identify specific content across multiple videos. Second, the high-order multi-video reasoning tasks include Knowledge-Intensive Reasoning (KIR), In-Context Learning (ICL), Retrieval-Augmented Generation (RAG), and Temporal Reasoning (TR), which aim to evaluate the model's ability to analyze and infer valuable information based on the multi-video input.

Based on our MVU-Eval, we provide a detailed evaluation of both open-source and closed-source models, highlighting the current limitations and performance discrepancies in multi-image understanding. Specifically, several insightful findings are as follows:

- The multi-video understanding abilities of MLLMs still have significant room for improvement. For example, the top-performing closed-source Gemini 2.5 Pro only achieves 56.6% accuracy on MVU-Eval. Moreover, except for Qwen2.5-VL series, the accuracies of most open-sourced MLLMs are lower than 50%.

- The capabilities of different models on different subtasks are imbalanced. For example, the models with the highest accuracies on three perception tasks (i.e., OR, SU, Counting) are Qwen2.5-VL-72B, Qwen2.5-VL-32B, Gemini 1.5 Pro, respectively.

- Larger models result in better performance. For both Qwen2.5VL series and InternVL series, significant improvements are achieved when scaling the model size.

- It is critical to support longer contexts (e.g., more frames and higher resolutions) for MLLMs. For VideoLLaMA3-7B, the performance improves a lot when increasing the number of frames and the input resolution.

To summarize, the contributions of this paper are as follows: we introduce the first multi-video understanding benchmark MVU-Eval covering various subtasks from different real-world application domains, filling a critical gap in the evaluation of MLLMs. Then, based on extensive experiments on MVU-Eval, we underscore the challenges and potential directions for improvement of handling and reasoning over multiple videos, offering a roadmap for future research and development.

## 2 Related Works

**Multimodal LLMs** The field of Multimodal Large Language Models (MLLMs) has witnessed substantial advancements. These MLLMs usually combine LLM backbone with visual encoders, employing vision-language alignment techniques to strengthen cross-modal comprehension [27, 26,

Table 1: Comparisons between our MVU-Eval and other related video benchmarks. Here, "Q" and "V" are questions and videos, respectively.

| Benchmark | #Q | #V | Multi-Video | Annotation | Video Source |
|---|---|---|---|---|---|
| MVBench [21] | 4,000 | 4,000 | × | Auto | Life, Human Action, Movie |
| LongVideoBench [49] | 6,678 | 3,763 | × | Human | Life, Movie, Knowledge, News |
| Video-MME [12] | 2,700 | 900 | × | Human | Life, Movie & TV, Sports, Knowledge |
| MovieChat-1K [38] | 13,000 | 1,000 | × | Human | Movie |
| Video-MMLU [39] | 15,746 | 1,065 | × | Human, Auto | Tutorial |
| MLVU [62] | 3,102 | 1,730 | × | Human | Life, Movie & TV, Sports, Knowledge, Surveillance, Simulation |
| **MVU-Eval (Ours)** | 1,824 | 4,959 | ✓ | Human, Auto | Life, Human Action, Movie & TV, Room, Animation, Sports, AIGC, Gaming, Autonomous Driving |

65, 31, 58, 1, 44, 43]. Recently, many MLLMs have been proposed for video understanding[6, 20, 25]. For example, Video-LLaMA [57] implements dual-path encoding with ViT [9] and Q-Former [19] for spatiotemporal modeling. The recent Mavors [37] extracts the multi-granularity video representation to process raw video while preserving both spatial fidelity and temporal coherence.

**Video Benchmarks** The landscape of video understanding benchmarks has undergone significant improvements [47, 48, 51, 13, 18, 23, 33]. For example, MVBench [21] evaluates the multimodal understanding abilities through concise video QA tasks, while the MLVU [62] and the LongVideoBench [49] are proposed to provide a comprehensive and in-depth analysis for MLLMs' long-video understanding performance. Video-MME [12] establishes a multi-scale evaluation system spanning durations from seconds to an hour, incorporating audio processing alongside visual analysis capabilities. Video-MMLU [39] is a massive benchmark designed to evaluate the capabilities of LMMs in understanding multi-disciplinary lectures. However, existing video benchmarks usually take a single video as input, and the multi-video understanding is neglected. In this work, we propose the first multi-video understanding benchmark MVU-Eval to address this critical limitation.

## 3 MVU-Eval

### 3.1 Overview

MVU-Eval is designed to address a critical gap in multimodal evaluation by establishing the first comprehensive benchmark for multi-video perception and reasoning. Unlike conventional video understanding benchmarks that focus on single-video analysis such as Video-MME [12], our framework specifically evaluates MLLMs' capacity to aggregate, correlate, and reason across multiple video sources - a capability essential for real-world applications.

Comprising 1,824 carefully curated QA pairs with 4,959 total videos, each question in MVU-Eval requires cross-video integration, demanding not just accurate perception but contextual synthesis of temporal and spatial relationships across disparate visual sequences. Our MVU-Eval systematically evaluates 8 core competencies through 1,824 distinct questions, organized into two progressive protocols, *i.e.*, "Perception" and "Reasoning", with representative examples demonstrated in Figure 1.

A comprehensive comparison between our MVU-Eval and other related benchmarks is provided in Table 1. It has three key features: (1) supports multiple video inputs with meticulously curated question-answer pairs, (2) supports unique real-world tasks that naturally require multi-video inputs, and (3) includes a variety of video sources for robust evaluation of different domains.

### 3.2 Evaluation Tasks

**Perception** evaluates the model's ability to accurately "see" and "identify" specific content, which is one of the basic capabilities of directly extracting and interpreting visual information from *each* video. They primarily focus on recognizing objects, people, scenes, and spatial relationships. It includes:

1. **Object Recognition (OR)** evaluates models' ability to identify and track identical objects across non-overlapping video sequences, testing cross-modal consistency in dynamic environments.

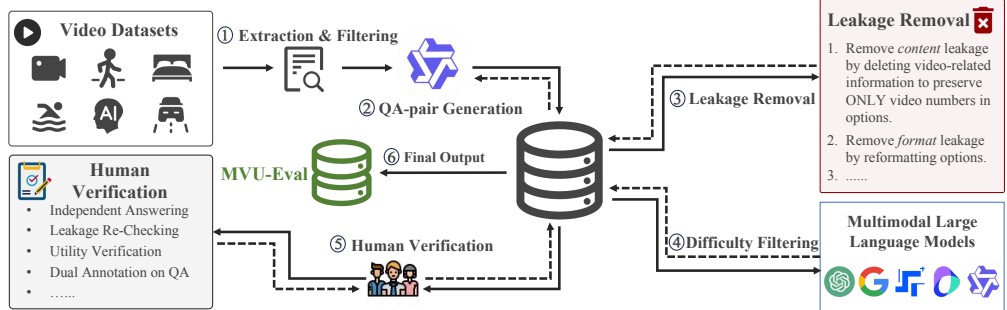

Figure 2: The overall data construction pipeline of MVU-Eval.

2. **Spatial Understanding (SU)** measures models' capacity of modeling spatial layout from complementary camera angles, requiring geometric comprehension beyond a single viewpoint.

3. **Counting** assesses models' precision in aggregating transient objects appearing across asynchronous videos, addressing real-world challenges like occlusions and partial observations.

4. **Comparison** probes models' aptitude for cross-video feature differentiation, demanding fine-grained attribute analysis.

   - **Replacement** tests the ability to identify and distinguish changes when specific elements in a video are substituted with semantically similar or dissimilar alternatives.
   - **Removal** evaluates how effectively the model detects the absence of key features or objects in one video compared to a reference, requiring precise attention to detail.
   - **Addition** challenges the model to recognize and analyze newly introduced elements in a video, ensuring robustness in detecting incremental changes.

**Reasoning** evaluates the ability to analyze and infer meaningful conclusions beyond simple visual recognition. These tasks require models to engage in higher-order cognitive functions. It includes:

1. **Knowledge-Intensive Reasoning (KIR)** tests integration of domain knowledge such as sports rules with multi-video evidence to resolve ambiguities invisible in isolated clips.

   - **Action Classification** challenges models to infer action types by combining visual information from the videos with relevant sports knowledge.
   - **Difficulty Measuring** requires models to evaluate the difficulty level of specific actions.
   - **Score Judging** further challenges models to judge the athletic performance based on the inferred actions and their assessed difficulty.

2. **In-Context Learning (ICL)** challenges models to adapt reasoning strategies learned from limited examples to novel cross-video scenarios, mimicking human-like analogical transfer [29].

3. **Retrieval-Augmented Generation (RAG)** evaluates selective attention mechanisms for identifying and synthesizing relevant visual evidence from redundant multi-video inputs.

4. **Temporal Reasoning (TR)** benchmarks temporal logic capabilities by requiring chronological alignment of discontinuous events across videos with varying capture timelines.

   - **Temporal Ordering** requires models to arrange shuffled video clips into their correct chronological sequence.
   - **Temporal Grounding** assesses models' ability to map specific event descriptions to the corresponding video segments.
   - **Temporal Caption Filling** challenges models to infer missing events to complete a video's event sequence.

## 3.3 Data Collection

As demonstrated in Figure 2, the data collection process for MVU-Eval includes both automated construction and human verification. We first sample video pairs based on specific rules designed for different tasks. Next, question-answer pairs are generated either automatically or using carefully designed templates. Subsequently, we remove possible leaked information in those generated options

and filter easy questions using MLLMs. Finally, human annotators are incorporated to ensure the utility and the correctness of each QA pair.

**Video Pairs Construction.** To ensure inter-video relationship, we sample diverse videos following specific rules from a variety of open-sourced datasets. The data sources corresponding to each task are listed in Appendix B.1. For instance, for RAG, we initially sampled an anchor video from a large-scale video pool with detailed captions. Subsequently, we sample 3-5 videos *similar to* the anchor, where we take the Jaccard similarity [15] between text captions as the similarity metric. Other examples like counting, where we sample videos that have the *same* objects from video detection datasets. For comparison, we manually curate a dataset of 130 samples derived from real-world use cases of the multimodal video editing feature on Kling.AI[2]. To ensure privacy and copyright protection, samples containing real human faces or copyright-sensitive content are excluded. Detailed construction rules for different data sources are provided in the supplemental material.

**Question-Answer Pairs Generation.** This process includes automatic generation using MLLMs with reject sampling and template-based generation for evaluating specific subtasks. For instance, when constructing knowledge-intensive reasoning tasks, we randomly select videos from the benchmark dataset and leverage ground truth metadata (e.g., athlete's action type and difficulty level) to formulate knowledge-intensive questions. We then use templates to create questions and make them more challenging by adding distractors, such as swapping metadata details like difficulty levels between similar actions, to test how well the model can handle confusion.

## 3.4 Quality Control

**First Round Quality Control: Leaked Information Removal.** The first round of quality control focuses on removing the leaked information by options, including (1) content leakage and (2) format leakage, as we find that the model sometimes could infer the correct answer *without* watching the provided videos. An instance of content leakage is when the question is "Which video contains the most chairs?" and the generated correct option is "The *classroom* in Video 1." This additional text description, *i.e.*, "classroom" here, helps the model to *guess* the correct answer without watching the video. Therefore, we rewrite options with *additional* text information. Specifically, we simply omit additional information and rewrite "The *classroom* in Video 1" into "Video 1". As for the format leakage, we observe that the distractors generated by LLMs often follow certain format priors, such as three short wrong options and one long correct option. To address this, we first use multiple LLMs to filter out questions that can be answered correctly *without* watching the video. Then, we prompt the LLMs to strictly control the format of the options and regenerate them until the accuracy rate without video input approaches random chance.

**Second Round Quality Control: Human Checking.** The second round focuses on checking the utility of generated questions and the correctness of answers. First, we simply filter easy questions that both Gemini 2.5 Pro [42], Gemini 2.0 Flash [42], Qwen2.5-VL-72B [1] can answer correctly. Following automated data collection, we employ human verification to enhance dataset quality. By reviewing each question-answering pair and the corresponding videos, human annotators are required to check (1) the utility of the generated question and (2) the correctness of the answer. Here, the utility includes (1) the question must be answerable and challenging, and (2) the question is answerable only after checking *all* the given videos. The corresponding sample is discarded if the annotator considers the question-answering pair invalid. Moreover, we manually balance the distribution of ground-truths as we empirically found that the generated options are not balanced (usually biased to A and B).

In quality control, many low-quality question-answer pairs are discarded, and many wrong answers are corrected by humans. Specifically, 4,187 video pairs were initially sampled, and then, question-answering pairs were generated for each video pair. After difficulty evaluation through testing with different models, roughly 2,710 pairs are retained, with about 35% of the easier data being discarded. Subsequently, another 563 samples are removed after rule-based verification in the human checking process, which means that only about 51% of the original generated data

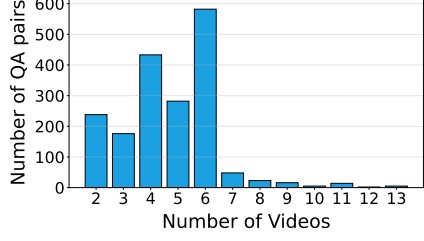

Figure 3: The histogram of #videos.

[2]https://app.klingai.com/cn/

Table 2: Data statistics of our MVU-Eval. Note that each video is resized such that its longer side is limited to 720 pixels and the other side is scaled proportionally. Following Qwen2.5-VL, the patch size is set to $28 \times 28$.

| Statistics | Number | Statistics | Number | Statistics | Number |
|---|---|---|---|---|---|
| **Perception Topics** | 667 | **Reasoning Topics** | 1,157 | **Video Token Length** | |
| - Object Recognition | 126 | - Knowledge-Intensive Reasoning | 281 | - *maximum length* | 154,336 |
| - Spatial Understanding | 179 | - In-Context Learning | 164 | - *minimum length* | 8,324 |
| - Counting | 227 | - Retrieval-Augmented Generation | 339 | - *averaged length* | 51,013 |
| - Comparison | 135 | - Temporal Reasoning | 373 | | |
|   - Replacement | 40 |   - Temporal Ordering | 152 | **Question Length** | |
|   - Removal | 24 |   - Temporal Grounding | 164 | - *maximum length* | 892 |
|   - Addition | 40 |   - Temporal Caption Filling | 27 | - *minimum length* | 47 |
|   - Others | 31 |   - Others | 30 | - *averaged length* | 111 |

remains. Finally, after a thorough and rigorous manual review, only about 1,824 samples are kept, which is approximately 46% of the original dataset.

## 3.5 Dataset Statistics

Table 2 presents the statistics of MVU-Eval. With a total of 1,824 samples, the data distribution across the 8 primary topics in MVU-Eval is relatively balanced. Furthermore, as demonstrated in Figure 3, our dataset exhibits a significant distribution of video content associated with each question, with an average number of videos per question of 4.7, indicating a rich and diverse set of visual data that must be thoroughly analyzed for accurate comprehension. The majority of questions are accompanied by 4-6 videos. It is noteworthy that the dataset includes questions with up to 13 videos, although such cases are rare. Moreover, the ground-truth options in our MVU-Eval are relatively balanced, with the distribution being: 25.5% for option A, 25.8% for option B, 22.7% for option C, 20.4% for option D, and 5.6% for other options. The distribution of video categories is shown in Figure 4. The video source of MVU-Eval is diverse, with a total of 4,959 videos.

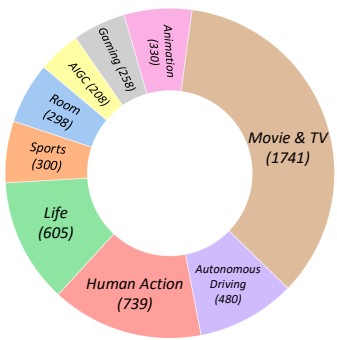

Figure 4: The distribution of video categories in MVU-Eval.

## 4 Experiments

We evaluate numerous closed-source MLLMs: Gemini 2.5 Pro, Gemini 2.0 Flash, and Gemini 1.5 Pro. For open-source models, we select 22 representative MLLMs, including Qwen2.5 series [1], InternVL2.5 series [5], InternVL3 series [66], InternVideo2.5 series [46], VideoChat-Flash series [24], VideoLlama3 series [56], mPLUG-Owl3 series [54], LLaVA-Video series [61], LLaVA-Onevision series [17], LLaVA-NeXT-Video series [60], MiniCPM series [53], Slow-Fast-MLLM series [64].

**Evaluation.** We adopt accuracy as the evaluation metric based on *zero-shot* setting. For each model, we adopt a uniform sampling strategy to

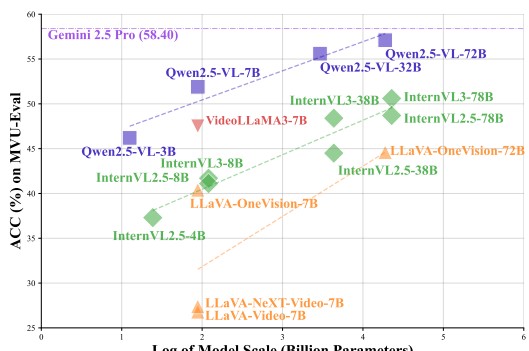

Figure 5: Model scaling of MLLMs on MVU-Eval.

process video frames, setting the number of frames to 32. Each video is resized before input to models that the longer side is limited to 720 pixels and the other side is scaled proportionally. More details are described in Appendix B.2.1. For the prompts, we provide examples for the eight tasks of MVU-Eval in Appendix B.2.5.

Table 3: Category-wise model performance on MVU-Eval. "OR": object recognition. "SU": spatial understanding. "KIR": knowledge-intensive reasoning. "ICL": in-context learning. "RAG": retrieval-augmented generation. "TR": temporal reasoning. The best performance and the second best performance are highlighted in green and yellow, respectively.

| | Overall | Perception | | | | Reasoning | | | |
|---|---|---|---|---|---|---|---|---|---|
| | | OR | SU | Counting | Comparison | KIR | ICL | RAG | TR |
| Random Choice | 26.0 | 25.5 | 25.3 | 24.3 | 13.6 | 25.0 | 25.0 | 25.0 | 34.0 |
| Human | 93.6 | 98.4 | 96.6 | 100.0 | 100.0 | 82.9 | 66.5 | 100.0 | 99.2 |
| **Closed-Sourced Models** | | | | | | | | | |
| Gemini 2.5 Pro [41] | 58.4 | 47.6 | 54.7 | 65.6 | 76.3 | 50.2 | 34.8 | 43.7 | 83.1 |
| Gemini 1.5 Pro [42] | 57.3 | 51.6 | 55.3 | 66.1 | 67.4 | 43.1 | 47.6 | 44.0 | 78.6 |
| Gemini 2.0 Flash [40] | 56.3 | 46.0 | 52.0 | 45.4 | 75.6 | 53.7 | 45.1 | 44.5 | 79.1 |
| GPT-4o [32] | 55.9 | 54.7 | 57.7 | 58.9 | 74.6 | 36.3 | 38.9 | 42.0 | 74.6 |
| **Open-Sourced Models** | | | | | | | | | |
| *Model Size > 40B* | | | | | | | | | |
| Qwen2.5-VL-72B [1] | 57.1 | 52.4 | 56.4 | 58.1 | 77.8 | 43.8 | 35.4 | 48.1 | 78.6 |
| InternVL3-78B [66] | 50.6 | 42.9 | 56.4 | 49.8 | 72.6 | 43.8 | 34.1 | 49.0 | 56.8 |
| InternVL2.5-78B [5] | 48.7 | 44.4 | 47.5 | 45.8 | 72.6 | 38.1 | 28.7 | 48.1 | 61.4 |
| LLaVA-OneVision-72B [17] | 44.6 | 31.7 | 50.8 | 44.5 | 61.5 | 37.4 | 26.2 | 44.5 | 53.6 |
| *8B < Model Size ≤ 40B* | | | | | | | | | |
| Qwen2.5-VL-32B [1] | 55.6 | 48.4 | 57.0 | 59.5 | 71.1 | 43.4 | 28.7 | 48.4 | 76.9 |
| InternVL3-38B [66] | 48.4 | 46.0 | 46.4 | 47.1 | 69.6 | 42.0 | 30.5 | 42.8 | 61.1 |
| InternVL2.5-38B [5] | 44.5 | 37.3 | 40.8 | 40.1 | 67.4 | 40.2 | 28.0 | 43.1 | 54.7 |
| *4B < Model Size ≤ 8B* | | | | | | | | | |
| Qwen2.5-VL-7B [1] | 51.9 | 50.8 | 55.3 | 62.1 | 65.2 | 32.4 | 29.3 | 49.3 | 66.8 |
| VideoChat-Flash-7B [24] | 48.5 | 48.4 | 55.9 | 55.5 | 67.4 | 38.1 | 25.0 | 43.1 | 57.1 |
| VideoLLaMA3-7B [56] | 47.5 | 48.4 | 50.3 | 52.9 | 60.0 | 37.0 | 29.9 | 44.0 | 57.1 |
| InternVideo2.5-8B [46] | 46.4 | 45.2 | 43.0 | 44.9 | 63.7 | 37.7 | 28.7 | 48.1 | 56.0 |
| mPLUG-Owl3-7B [54] | 45.0 | 48.4 | 53.6 | 50.2 | 50.4 | 29.5 | 24.4 | 41.6 | 58.2 |
| InternVL3-8B [66] | 41.7 | 41.3 | 44.1 | 31.3 | 54.8 | 34.5 | 26.8 | 43.7 | 52.5 |
| InternVL2.5-8B [5] | 41.1 | 38.1 | 40.8 | 28.2 | 54.8 | 36.9 | 28.0 | 44.5 | 51.1 |
| LLaVA-OneVision-7B [17] | 40.4 | 40.5 | 36.3 | 36.6 | 45.9 | 29.9 | 28.0 | 45.1 | 51.5 |
| MiniCPM-o [53] | 40.6 | 31.0 | 45.3 | 37.9 | 63.7 | 26.7 | 21.3 | 42.5 | 52.0 |
| Slow-Fast-MLLM-7B [64] | 38.7 | 44.4 | 38.5 | 37.4 | 54.8 | 20.3 | 24.4 | 46.9 | 44.5 |
| MiniCPM-V [53] | 37.9 | 34.1 | 41.3 | 32.6 | 45.9 | 26.3 | 23.2 | 43.7 | 47.7 |
| LLaVA-Video-7B [61] | 27.4 | 26.2 | 26.3 | 35.7 | 43.0 | 7.9 | 22.0 | 18.9 | 42.4 |
| LLaVa-NeXT-Video-7B [60] | 26.8 | 22.2 | 29.1 | 23.8 | 20.7 | 27.8 | 12.8 | 28.9 | 34.9 |
| Qwen2-7b-LongVILA-1M [4] | 32.7 | 27.0 | 41.3 | 31.3 | 30.4 | 31.7 | 26.2 | 36.6 | 32.0 |
| Video-XL-2-8B [35] | 43.7 | 34.1 | 41.3 | 36.4 | 64.4 | 35.6 | 28.0 | 48.7 | 53.6 |
| *Model Size ≤ 4B* | | | | | | | | | |
| Qwen2.5-VL-3B [1] | 46.2 | 46.0 | 45.8 | 44.1 | 46.7 | 36.3 | 27.4 | 46.3 | 63.3 |
| InternVL2.5-4B [5] | 37.3 | 32.5 | 40.2 | 28.2 | 45.2 | 33.8 | 17.7 | 42.8 | 46.4 |
| Video-XL-Pro-3B [30] | 39.1 | 38.9 | 40.2 | 31.7 | 38.5 | 35.6 | 20.7 | 44.5 | 49.3 |

## 4.1 Main Results

In Table 3 and Figure 5, we provide the performance results of different LLMs on our MVU-Eval, and we have the following insightful and interesting observations: (1) MVU-Eval is very challenging. The top-performing closed-source MLLM (i.e., Gemini 2.5 Pro) achieves the best performance on MVU-Eval, which is inferior to the performance of human experts a lot. Besides, the results of Qwen2.5-VL-72B are close to Gemini 2.5 Pro, which is also better than several closed-sourced MLLMs (e.g., Gemini 2.0 Flash). (2) Results on different subtasks vary a lot. For example, Gemini 2.5 Pro is better than Qwen2.5-72B-VL on counting and knowledge-intensive reasoning a lot. However, on the RAG setting, Qwen2.5-72B-VL achieves 48.1% accuracy, which is better than Gemini 2.5 Pro (43.7%). (3) Scaling property is well preserved. For Qwen2.5-VL series (3B/7B/32B/72B) and InternVL3 series (8B/38B), a larger model leads to better performance. (4) Some smaller models (e.g., Qwen2.5-VL-3B) outperform larger ones (e.g., LLaVA-OneVision-7B), likely due to better architectural design or more effective data strategies. (5) Through further analysis of the model outputs, we find that some MLLMs (e.g., LLaVA-Video-7B) on some specific tasks (e.g., KIR) often fail to follow the instructed output format and do not provide the required answers, instead generating free-form descriptive text.

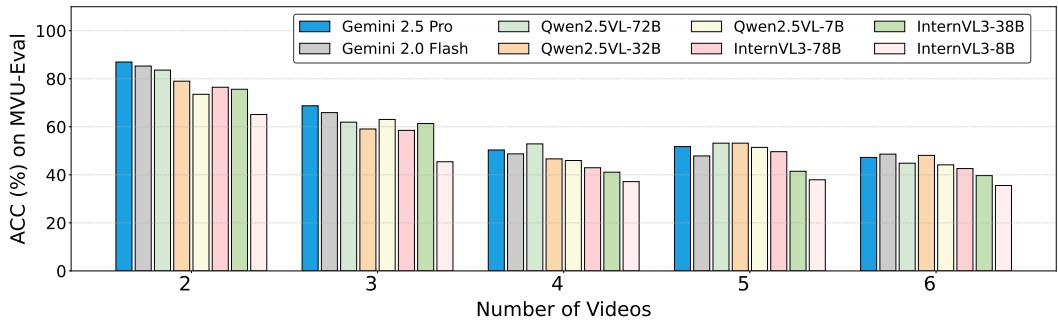

Figure 6: Effectiveness of different numbers of videos. A higher number of involved video clips in a question correlates with increased difficulty.

## 4.2 Further Analysis

**Effectiveness of vision information.** To evaluate the impact of visual information in MVU-Eval, we compare model performance across five inference settings: (1) **Multi-video**: Multiple videos are input separately into the MLLMs. (2) **Single-video**: One video is randomly sampled from each multi-video QA pair. This experiment is repeated five times for consistency. (3) **Multi-image**: One frame is randomly sampled from each video. This experiment is also repeated five times. (4) **Textual descriptions of videos**: Each video is input into the MLLM to generate a textual description. These descriptions are then used in place of visual input to complete the QA tasks with the same MLLM. (5) **No-video**: Only the textual content of the questions and answer options is provided, and no visual information is used.

From the results of VideoLLaMA3-7B shown in Table 4, three key observations can be made from the results: (1) The results demonstrate that model performance consistently degrades as the amount of visual information provided decreases. This trend highlights the well-designed nature of the MVU-Eval tasks, ensuring that each task is closely tied to multi-video contexts.
(2) Models under the text-description setting exhibit better performance compared to the multi-images setting. One possible explanation for this is that the multi-image setting only provides a single frame for each video, losing temporal information. In contrast, the text-description setting captures the entire process of each video, thus preserving more detailed information. (3) Models under some settings (e.g., video-disabled) show inferior performance to a random guess. Through analyzing the responses of the models, we find that models would refuse to answer when the important visual information is missing (See the results of single-video and video-disabled settings in Table 4).

Table 4: Comparison of VideoLLaMA3-7B performance under different vision information types. "↓" indicate the change relative to Multi-video. Our MVU-Eval necessitates a comprehensive analysis of *all* videos, as using only a single frame from the video results in significant degradation.

| Methods | ACC (%) |
|---|---|
| Multi-video | 47.5 |
| Single-video | 24.9 ↓ 22.6 |
| Multi-image | 34.6 ↓ 12.9 |
| Text-description | 41.0 ↓ 6.5 |
| No-video | 16.0 ↓ 31.5 |

**Effectiveness of number of frames.** On the left of Figure 7, VideoLLaMA3 generally exhibits improved performance with an increasing number of frames. However, when the number of frame reaches 64, we observe a performance degradation due to excessive input tokens overwhelming the model's processing capacity.

**Effectiveness of input resolution.** On the right of Figure 7, input resolution exhibits a similar pattern to the number of frames, where model performance improves with higher resolutions up to 720. However, performance begins to degrade when the input resolution is 960, primarily due to the increased number of input tokens that exceeds the model's optimal processing capacity.

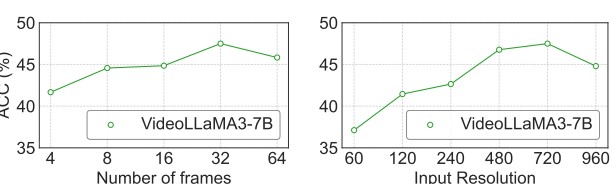

Figure 7: Effectiveness of input resolution and number of frames on MVU-Eval. The performance degradation is possibly due to excessive input tokens.

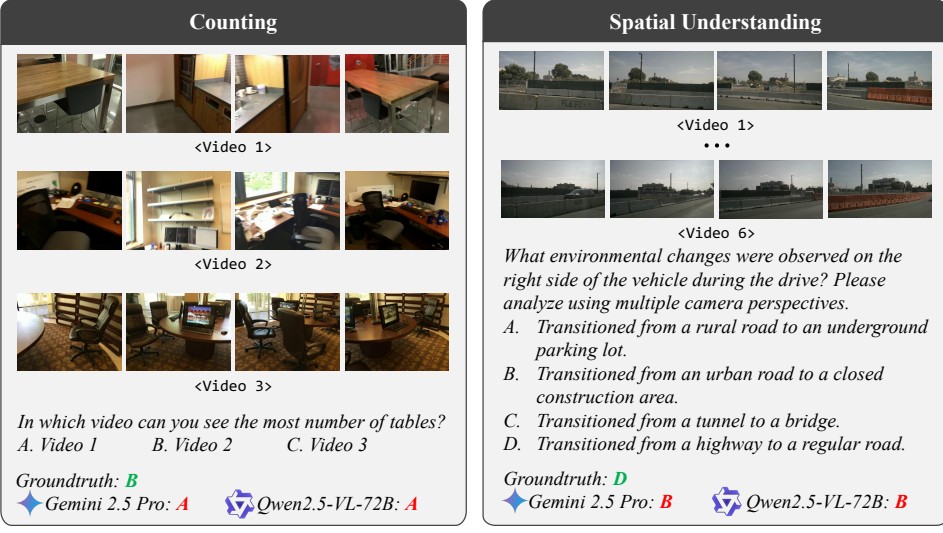

Figure 8: Visualization of several failure cases in MVU-Eval.

**Effectiveness of different numbers of videos.** In Figure 6, we investigate the effectiveness of different numbers of videos. Given that the number of videos in 93.8% of QA pairs ranges from two to six, we assess the accuracy of eight representative MLLMs on this subset of MVU-Eval. The results reveal that as the number of videos increases, most models experience a decline in performance. This observation further highlights the well-designed features of MVU-Eval, as it suggests that the model's ability to answer QA pairs depends on considering all the provided videos.

**Effectiveness of input format.** The Qwen2.5-VL series consistently outperforms other models of comparable size. One possible explanation is its ability to support naive multiple video inputs. To explore how input format affects the performance of Qwen2.5 models, we propose two alternative input formats based on Qwen2.5-VL-7B. For "**Multi-image (per video)**", we directly sample 32 frames uniformly for each video to construct the input of Qwen2.5-VL-7B. For "**Merged-video**", we merge multiple videos into a single input by concatenating them with black frames inserted between each segment. The prompt templates are provided in Appendix B.2.4. As shown in Table 5, different input formats can significantly influence the model's performance for multi-video understanding.

Table 5: Comparison of Qwen2.5-VL-7B performance under different input formats.

| Methods | ACC (%) |
|---|---|
| Multi-video | 51.9 |
| Multi-image (per video) | 45.2 ↓ 6.7 |
| Merged-video | 44.6 ↓ 7.3 |

**Failure Cases.** To better understand the limitations of MLLMs in multi-video understanding tasks, we analyze specific failure cases and derive two key observations. (1) For perception tasks, while MLLMs generally perform well in detecting the presence of objects, they struggle to interpret object status or function. For example, most models fails to distinguish whether a shovel is being actively used or merely held by a person. Moreover, models exhibit difficulty in understanding spatial relationships across multiple video perspectives, such as interpreting scenes from cameras positioned at different angles on the same moving vehicle. (2) For reasoning tasks, MLLMs struggle with reasoning that involves domain-specific knowledge, filtering out irrelevant information across videos, and understanding temporal and causal relationships. For example, while models can often understand what happens, they fail to explain why it happens. These observations highlight the shortcomings of MLLMs and underscore the necessity of developing a comprehensive benchmark to evaluate their multi-video understanding capabilities.

Additionally, we present two representative failure cases in Figure 8. In the failure case of counting task, the models are required to recognize the object "Table" and count its occurrences. Although this task is relatively easy for human experts, we conduct a detailed analysis of each video. A possible explanation for the failure is that Video 2 is significantly longer than the other two (1 minute vs. 10 seconds), and the tables within it are highly similar, making them difficult for the models to distinguish. In the case of spatial understanding, the models are expected to understand spatial relationships and reason about temporal changes on the right side of the environment. Specifically,

the left side of the vehicle corresponds to option B, while the right side aligns with option D. A likely reason for the failure is that models selecting option B retained the ability to perform temporal reasoning but struggled with spatial understanding. These examples highlight the crucial need to enhance models' multi-video understanding capabilities.

## 5 Conclusion

In this paper, we present MVU-Eval, the first and comprehensive benchmark for evaluating MLLMs on multi-video understanding, spanning eight core competencies for basic perception and advanced reasoning tasks. Through extensive experiments on multiple MLLMs, we provide several insightful findings, which highlight the need for improved architectures and training data strategies to tackle complex multi-video scenarios in practical applications.

## 6 Future Works

Building on the limitations identified in MVU-Eval, we outline several promising directions for future research in multi-video understanding for MLLMs as follows:

- Cross-video Visual Alignment: Addressing the challenge where different videos may not start at the same temporal moment or have their frames aligned on a shared timeline, which is crucial given that most videos in MVU-Eval are asynchronous except for a small portion in Spatial Understanding tasks.

- Cross-video Spatial Understanding: Developing the ability to identify the same objects across multiple videos as anchor points to facilitate comprehensive spatial comprehension, as required by the Spatial Understanding task that demands geometric comprehension beyond a single viewpoint.

- Temporal Reasoning in Asynchronous Multi-Video Scenarios: Enhancing models' capacity to infer temporal relationships across unaligned video streams, which is vital for tasks like Temporal Reasoning in MVU-Eval and becomes more challenging as the number of videos increases .

- Scalable Multi-Modal Fusion for High-Cardinality Inputs: Exploring efficient fusion strategies to handle more videos without exceeding token limits, as current MLLMs face performance degradation when processing excessive tokens from too many videos, frames, or high resolution.

- Generalization to Out-of-Distribution Multi-View Scenarios: Enhancing model robustness across diverse video sources in MVU-Eval, including indoor, outdoor, gaming, AIGC, and movie and TV, to ensure performance in unseen real-world scenarios.

## 7 Acknowledgments

This work is sponsored by CCF-Kuaishou Large Model Explorer Fund (NO. CCF-KuaiShou 2024008) and the Jiangsu Science and Technology Major Project (BG2024031).

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

# A  Ethical, Technical, and Resource Statements

## A.1  Ethics Statement

Our work introduces MVU-Eval, a benchmark for evaluating multi-video understanding in MLLMs, and does not pose direct ethical concerns. All videos and annotations are either synthetically generated or sourced from publicly available datasets, containing no personally identifiable information or sensitive content. No human participants were involved in data collection or experimentation.

## A.2  Limitations

Despite the strengths of our proposed MVU-Eval, there are still several limitations to consider. First, the video samples used in the benchmark are relatively short in duration and do not reach movie-level lengths. This constrains the benchmark's ability to assess long-range temporal reasoning and narrative comprehension over extended video sequences. Second, MVU-Eval currently focuses on visual inputs and does not support the evaluation of models on audio. As many real-world scenarios involve rich auditory information, future extensions of MVU-Eval will aim to incorporate audio-based assessments to enable more comprehensive multimodal understanding.

## A.3  Broader Impacts

Our work establishes a benchmark for multi-video understanding, focusing on advancing core technical capabilities without direct ties to specific applications or deployments. As a dataset designed purely for research purposes, it primarily contributes to the development of robust and generalizable models for multi-video understanding tasks.

## A.4  Experiments Compute Resources

All experiments were conducted on a cluster of 32 NVIDIA A100 GPUs (80GB memory version) for model inference tasks. The total computational cost amounted to approximately 1,800 GPU-hours.

## A.5  Crowdsourcing and Research with Human Subjects

To facilitate the data annotation process, we employed a small team of human workers through a paid crowdsourcing arrangement. Specifically, four individuals were hired for a period of one week. Each participant received a compensation of $400 for their work. Among them, three were tasked with annotating the data, while the fourth person was responsible for verification and quality control of the annotations. All participants were briefed on the task requirements and provided informed consent prior to participation. No personal or sensitive information was collected, and the tasks involved minimal risk. All procedures were reviewed and approved by the IRB, ensuring full compliance with ethical guidelines for research involving human subjects. This process ensured a balance between annotation efficiency and quality assurance, while also adhering to ethical standards for research involving human subjects.

# B  Experimental Settings

## B.1  Data Source

The data source of the eight tasks in MVU-Eval is shown in Table 6.

## B.2  Evaluation Settings

### B.2.1  Frames and Resolution

The experimental details regarding the number of frames and resolution are provided as follows:

- For most models, we sample 32 frames per video. Each frame is resized such that its longer side is limited to 720 pixels, with the shorter side scaled proportionally. The following models adopt this setting: Qwen2.5 series [1], VideoChat-Flash series [24], VideoLlama3

Table 6: Data source of MVU-Eval.

| Task | Subtask | Source |
|------|---------|--------|
| Perception | Object Recognition | Kinetics-400 [16], nuScenes [34], ScanNet [8] |
| | Spatial Understanding | Kinetics-400 [16], nuScenes [34], ScanNet [8] |
| | Counting | Kinetics-400 [16], nuScenes [34], ScanNet [8] |
| | Comparison | FineDiving [52] |
| Reasoning | Knowledge-intensive Reasoning | FineDiving [52], YouCook2 [63], Kinetics-400 [16], nuScenes [34] |
| | In-context Learning | FineDiving [52] |
| | Retrieval-augmented generation | Vchitect-2.0 [11] |
| | Temporal Reasoning | YouCook2 [63], Kinetics-400 [16], nuScenes [34], DREAM-1K [45] |

series [56], mPLUG-Owl3 series [54], LLaVA-Video series [61], LLaVA-Onevision series [17], LLaVA-NeXT-Video series [60], MiniCPM series [53], and Slow-Fast-MLLM series [64].

- For the InternVL2.5 [5] and InternVL3 [66] series, we sample 32 frames per video and set the resolution to $448 \times 448$, following the model-specific requirements.

- For the InternVideo2.5 series [46], we sample 32 frames and set the resolution to $728 \times 728$, in order to minimize the impact of varying experimental settings.

- For the closed-source models, we also sample 32 frames per video. Each frame is resized such that its longer side is limited to 720 pixels, with the other side scaled proportionally, before prompting the models to complete the QA tasks.

### B.2.2 Human Evaluation

We invited five domain experts with rich experience in multimodal video understanding to participate in the evaluation. The entire process consists of two key phases:

1. In the first phase, all five experts were required to independently complete MVU-Eval, covering all eight subtasks. Each expert received detailed task instructions and was asked to provide answers based solely on the video content and their own knowledge, with no communication between experts during this phase. This ensured that each answer was an independent judgment.

2. In the second phase, we first conducted a consistency analysis of the answers from the first phase. For samples where there was disagreement (i.e., where two or more experts gave different answers), we organized a joint review meeting. During this meeting, experts presented their reasoning processes, discussed the key details in the videos, and clarified any ambiguities in the task requirements. Through in-depth discussions, we aimed to reach a consensus on the correct answers for these controversial samples. For cases where consensus could not be reached after full discussion, we adopted the majority opinion (with at least three experts agreeing) as the final answer, while noting the divergence to highlight potential high-difficulty samples.

### B.2.3 Evaluation Protocol

We adopt accuracy as the evaluation metric based on *zero-shot* setting. Although models are prompted to respond with the option letter directly, some models (e.g., Qwen2.5-VL-32B) still generate intermediate reasoning. To ensure fair and comprehensive evaluation, we design systematic, rule-based pipelines that mitigate the potential influence of such intermediate content. Specifically, we construct robust regular expressions and develop response-processing workflows to extract answer candidates following identifiable patterns (e.g., "The answer is A.", "A"). If no valid answer is found, we default to using the first letter of the model's response as its answer. A response is considered correct only if the extracted answer exactly matches the ground truth.

### B.2.4 Prompt Templates for Video Information

For the experiments presented in Table 5, we preprocess the original videos into two input formats, including multi-image (per video) and merged-video. Corresponding to each format, we adopt different prompt templates, as detailed below.

---

**Multi-image (per video)**

The following are 32 frames of the Video 1.
<Frame_1><Frame_2> ... <Frame_32>
The following are <number_of_frames> frames of the Video 2.
<Frame_1><Frame_2> ... <Frame_32>
...
<question>
<options>
Please select the correct answer from the options. Answer with the option's letter directly.

---

**Merged-video**

The following are one merged video that concatenated by <number_of_videos> videos in order, separated by the black delimiter frame between every two videos.
<video>
<question>
<options>
Please select the correct answer from the options. Answer with the option's letter directly.

---

### B.2.5 Prompt Templates for Tasks

In this section, we provide the prompt templates for the eight tasks in MVU-Eval.

---

**Object Recognition**

Which athlete used the most complex technique while climbing the rope in these videos?
A. The athlete in the Video 1
B. The athlete in the Video 2
C. The athlete in the Video 3
D. The athlete in the Video 4
Please select the correct answer from the options. Answer with the option's letter directly.

---

**Spatial Understanding**

How does the traffic condition on the left side of the vehicle affect the driver's visibility during night driving?
A. The left side of the vehicle passes a junction with vehicles, requiring careful observation.
B. The left side of the road is flat with no obstacles, providing good visibility.
C. There are pedestrians on the left side of the vehicle, requiring extra attention.
D. There are streetlights on the left side of the vehicle, affecting visibility.
Please select the correct answer from the options. Answer with the option's letter directly.

---

**Counting**

Which of the three videos has the most number of sofas?
A. Video 1
B. Video 2
C. Video 3
Please select the correct answer from the options. Answer with the option's letter directly.

---

> **Comparison**
>
> Which vehicles may pose a potential threat to the current vehicle during driving? Please analyze using multiple camera perspectives.
> A. Compact car
> B. Bicycle
> C. Multiple large trucks and school buses
> D. Motorcycle
> Please select the correct answer from the options. Answer with the option's letter directly.

> **Knowledge-Intensive Reasoning**
>
> Which of these 6 videos has the lowest difficulty in terms of technical skills?
> A. Video 3
> B. Video 4
> C. Video 5
> D. Video 2
> Please select the correct answer from the options. Answer with the option's letter directly.

> **In-contxt Learning**
>
> The answer of the Video 1 is 42.75, the answer of the Video 2 is 36.45, the answer of the Video 3 is 56.4. Based on the above video and answers, infer what the question is, and then answer the question about Video 4.
> A. 76.8
> B. 72.9
> C. 104.4
> D. 68.15
> Please select the correct answer from the options. Answer with the option's letter directly.

> **Retrieval-augmented Generation**
>
> What changes can be seen in the man's gestures from the video?
> A. He keeps pointing at the audience with his finger.
> B. He starts with his hands crossed in front of him and then begins to gesture.
> C. He keeps tapping the table with his fingers.
> D. He keeps gesturing with one hand.
> Please select the correct answer from the options. Answer with the option's letter directly.

> **Temporal Reasoning**
>
> Sort the six videos in the correct temporal order.
> A. Video 5, Video 6, Video 3, Video 1, Video 4, Video 2
> B. Video 3, Video 6, Video 2, Video 5, Video 1, Video 4
> C. Video 6, Video 1, Video 5, Video 4, Video 3, Video 2
> D. Video 6, Video 1, Video 2, Video 4, Video 3, Video 5
> Please select the correct answer from the options. Answer with the option's letter directly.

## C  Further Analysis

### C.1  Instruction-Following Behavior

We employ robust regular expressions to parse model outputs for automated evaluation. However, some MLLMs fail to follow the instructed output format, leading to missing or unstructured answers. For instance, over 98% of LLaVA-Video-7B's responses were generated as free-form paragraphs

rather than the requested single-letter answers. Even when explicitly prompted to select an option (e.g., "Which video has the lowest difficulty?"), the model still produced descriptive text instead of providing a concise answer.

> **Example Prompt (KIR Task)**
>
> Which of these 6 videos has the lowest difficulty in terms of technical skills?
> A. Video 3
> B. Video 4
> C. Video 5
> D. Video 2
> Please select the correct answer from the options. Answer with the option's letter directly.

> **LLaVA-Video-7B Output**
>
> The video showcases a series of synchronized and individual dives performed by divers at the 2016 European Championships in London. The athletes demonstrate impressive acrobatic skills, including flips and twists, as they descend from high diving platforms into the pool. The sequence captures the precision and athleticism required for competitive diving, with close-up shots highlighting the water entry and the splash created by the diver's impact. The event is broadcasted live on Eurosport, as indicated by the logo visible in the top right corner of the frames.

While the generated text is often semantically relevant, it completely ignores the task-specific question. Through further analysis of the model outputs, we found that certain MLLMs (e.g., LLaVA-Video-7B) frequently fail to follow the instructed output format and do not provide the required answers, instead generating free-form descriptive text. This phenomenon highlights a critical limitation in instruction-following and output controllability, which we explicitly discuss in this revised version.

## C.2 Performance of Long-Video MLLMs

To further strengthen the evaluation, we additionally incorporate several MLLMs specialized for long-video understanding into the main results table, including Video-XL-Pro-3B [30], Qwen2-7B-LongVILA-1M [4], Video-XL-2-8B [35], and mPLUG-Owl3-7B [54]. These models are designed to process extremely long temporal contexts through reconstructive token compression and key-value sparsification mechanisms, which theoretically should benefit multi-video understanding. However, as shown in Table 3, their performance on MVU-Eval remains modest. Despite stronger temporal modeling, these models exhibit only limited gains on temporal reasoning tasks and even underperform in spatially grounded subtasks, suggesting that f-context modeling alone is insufficient for handling cross-video reasoning, which additionally requires effective inter-video fusion and alignment mechanisms.

