# MVU-Eval: Towards Multi-Video Understanding Evaluation for Multimodal LLMs (Supplementary Material)

**Tianhao Peng**[*1,4], **Haochen Wang**[*2], **Yuanxing Zhang**[*3], **Zekun Wang**[3], **Zili Wang**[4],
**Ge Zhang**[4], **Jian Yang**[4], **Shihao Li**[1], **Yanghai Wang**[1], **Xintao Wang**[3], **Houyi Li**[4],
**Wei Ji**[1], **Pengfei Wan**[3], **Wenhao Huang**[4], **Zhaoxiang Zhang**[1,2], **Jiaheng Liu**[†,1]
[1]Nanjing University, [2]CASIA, [3]Kuaishou Technology, [4]M-A-P

## 1 Detailed Construction Pipeline

In this section, we introduce the construction pipeline for generating MVU-Eval QA pairs based on each data source.

### 1.1 Kinetics-400 & nuScenes & ScanNet & Vchitect-2.0

For videos sampled from Kinetics-400 [6], nuScenes [2], ScanNet [3], and Vchitect-2.0 [4], we construct a semi-automatic pipeline. Specifically, after constructing video samples, we prompt the Qwen2.5-VL-72B-Instruct [1] to generate multiple-choice questions and answers, with videos and their labels (if possible) as inputs. These questions include: (1) **Object Recognition**, (2) **Spatial Understanding**, (3) **Counting**, (4) **Knowledge-intensive Reasoning**, and (5) **Temporal Reasoning**. These generated questions, answers, and candidate choices are manually checked by humans. Pipelines for constructing video pairs are slightly different across datasets.

**Kinetics-400.** By default, 2-6 videos are randomly sampled, regardless of their labels. To generate challenge questions, we additionally sample video pairs that belong to the exact *same* category.

**nuScenes.** Each video pair includes 6 videos from the different camera perspectives, *i.e.*, left-front, front, right-front, left-back, back, and right-back.

**ScanNet.** We randomly sample 2-6 videos for each question. We take detection labels as inputs with a probability of 50% as we find that generated questions are usually about counting when taking detection labels as inputs. Therefore, more diverse questions are generated without detection labels.

**Vchitect-2.0.** We first randomly sample an anchor video, and make the LMM to generate fine-grained questions according this video. Subsequently, we sample 3-5 videos *similar to* the anchor, where we take the Jaccard similarity [5] between text captions as the similarity metric.

### 1.2 FineDiving

For the data source of FineDiving [11], we develop a systematic approach to generate question-answer pairs for Knowledge-intensive Reasoning and In-context Learning tasks. Our process leverages the rich metadata provided for each diving video clip, which includes the difficulty coefficient of the dive, the type of action performed, and the score received. For **Knowledge-intensive Reasoning** tasks, we randomly sample six videos from the dataset. Using the metadata of these videos, we formulate questions with definitive correct answers, such as "In these 6 videos, which videos have the same difficulty coefficient for the athletes?" The correct answer was derive directly from the metadata. We then generate distractor options that are similar in format but incorrect in content to increase the task's difficulty. For **In-context Learning** tasks, we focus on two key metadata elements, including action difficulty and score. We randomly sample 4 videos for each question. Following a template, we

provide information about the difficulty/score of the first three videos in the question text. The correct answer option was the difficulty/score of the fourth video. To create distractors, we randomly sample difficulty/score information from 3 other videos in the database. This approach allow us to create a challenging benchmark that tests both the model's ability to reason using domain-specific knowledge and its capacity to learn and apply patterns from context. By utilizing the inherent relationships within the diving metadata, we ensure that the questions were both relevant to the sport and require deep understanding of the video content and associate information.

## 1.3 YouCook2

For the YouCook2 [12] dataset, which consists of long videos demonstrating complete recipe preparations, we develop tasks to test Knowledge-intensive Reasoning and Temporal Reasoning. We leverage the dataset's metadata, where each video represents a recipe and is composed of key steps. First, we segment the videos into shorter clips, each representing an essential cooking step, based on the original dataset labels. For the **Knowledge-intensive Reasoning** task, we present all clips of a recipe to the model in their original sequence. The question asks "Based on the <video_num> videos, infer the dish being made and describe the cooking process." The correct answer is derived from the dataset's step-by-step descriptions. To create distractors, we use Qwen2.5-72B-Instruct [8] to generate incorrect but plausible options. For the **Temporal Reasoning** task, we shuffle the clips of each recipe. The correct answer is the accurate sequence of steps, and distractors are created by randomly reordering the steps. This approach creates a benchmark that tests the model's ability to understand video segments and reason through complex processes, ensuring questions are based on real-world cooking scenarios for practical evaluation.

## 1.4 DREAM-1K

**Comparison.** In this task, we aim to assess the ability of models to discern differences between pairs of similar videos and to generate the minimal operations required to transform a source video into a target video. The video editing task is categorized into three sub-tasks based on the type of editing operation: (1) Replacement, (2) Removal, and (3) Addition. We manually curate a dataset of 130 samples derived from real-world use cases of the multimodal video editing feature on Kling.AI[1], comprising 50 samples for Replacement, 30 for Removal, and 50 for Addition. To ensure privacy and copyright protection, samples containing real human faces or copyright-sensitive content are excluded. Each selected sample consists of a source video, a ground-truth user prompt specifying the video editing instructions, and the corresponding edited target video. To create candidate options for each sample, we first employ Mavors [7], an advanced 7B-size video LLM, to generate captions for both the source and target videos, and subsequently, we prompt Qwen2.5-32B-Instruct [8] to generate nine negative options based on the video captions by altering attributes such as object, action, quantity, position, or the scope of changes (*e.g.*, global *v.s.* local) in the ground-truth user prompt. These generated negative options are then manually reviewed and filtered to ensure they are incorrect. The resulting dataset contains an average of 9.82 options per sample.

**Temporal Reasoning.** To evaluate models' capabilities in understanding temporal dependencies, narrative integrity, and event grounding within videos, we propose three distinct tasks: (1) Temporal Ordering, which requires models to arrange shuffled video clips into their correct chronological sequence; (2) Temporal Grounding, which assesses models' ability to map specific event descriptions to the corresponding video segments; and (3) Temporal Caption Filling, which challenges models to infer missing events to complete a video's event sequence. We construct the datasets for these tasks using DREAM-1K [9], selected for its rich multi-event video content.

The data pipeline for all three tasks begins with a shared four-stage process—(1) video segmentation, (2) clip captioning, (3) event merging and scoring, and (4) data filtering—followed by task-specific steps to construct the final datasets for temporal ordering, temporal grounding, and temporal caption filling. In the first stage, we employ PySceneDetect[2] to segment videos into clips using a threshold of 27.0. Subsequently, these clips are captioned using Mavors [7], with a focus on generating overall descriptions of each clip's content. As scene-based segmentation may not align perfectly with event boundaries, we utilize Qwen2.5-32B-Instruct[8] in the third stage to merge consecutive

---

[1] https://app.klingai.com/cn/
[2] https://github.com/Breakthrough/PySceneDetect

clip descriptions into events based on their semantic similarity, employing elaborate In-Context Learning and Chain-of-Thought [10] prompting techniques. The ground-truth event descriptions from DREAM-1K provide contextual guidance for the event merging and scoring process. Concurrently, the model evaluates the temporal structure of the merged events by assigning three metrics, each scored from 0 to 10: (1) Sequential Coherence, which measures the logical coherence of the event sequence and the necessity of maintaining a specific order; (2) Logical Predictability, which evaluates whether earlier events enable accurate prediction of subsequent events; and (3) Event Completeness, which assesses the impact of event missing on the narrative integrity of the sequence. Finally, we filter the dataset by excluding samples with a sequential coherence score below 6, a logical predictability score below 5, an event completeness score below 7, a number of events below 2 (below 3 for the Temporal Caption Filling task), and where non-consecutive clips are merged into an event.

Using the filtered event data, we construct datasets for the three tasks through task-specific procedures. For Temporal Ordering, we randomly shuffle the order of event video clips and generate incorrect orderings as negative options. For Temporal Grounding, we select an event description and generate multiple-choice options with the correct clip index and randomly sampled incorrect clip indices. For Temporal Caption Filling, we prompt Qwen2.5-32B-Instruct [8] to mask an event description and generate multiple-choice options with the correct description and plausible but incorrect alternatives. All event descriptions except for the masked ones will be replaced with the corresponding video clips. Finally, all data are manually reviewed to ensure (1) no multiple answers exist; (2) consistency between event descriptions and their corresponding video clips; and (3) accuracy of the options. We obtain 95, 200, and 33 samples for Temporal Ordering, Temporal Grounding, and Temporal Caption Filling tasks, respectively.