# OpenReview forum: "MVU-Eval: Towards Multi-Video Understanding Evaluation for Multimodal LLMs"
_NeurIPS.cc/2025/Datasets_and_Benchmarks_Track — NeurIPS 2025 Datasets and Benchmarks Track poster_

### Official Review · Reviewer_KGwg · 2025-06-29

**Rating:** 4
**Confidence:** 4

**Summary:**

This paper introduces MVU-Eval, a multi-video benchmark for multimodal LLMs. It covers both perception and reasoning tasks. The authors evaluate many video LLMs and provide some analysis of their performance.

**Dataset Code Accessibility:**

Yes

**Dataset Code Comments:**

The huggingface repo looks complete to me

**Ethical Considerations:**

No, there are no or only very minor ethics concerns

**Final Justification:**

I read the author's rebuttal, but still have concerns about the video sources. As a general benchmark for videos, it should have more diverse videos, especially videos from YouTube on various topics. I would maintain my original score.

**Limitations Weaknesses:**

First, I didn't find where the authors explain how the QA pairs are generated, what are the videos sources, and how the videos are selected, which is very important for reviewing the quality of the benchmark and the quality of this paper.

Second, the authors do not explain  how the audio information is processed, which is important in video understanding. It is totally OK If the benchmark is designed for not using audio, but be very clear about that would be helpful.

Finally, the selected task is limited, for example, there is only object recognition for recognition, but there could be relation/attribute/action recognition and so on. For reasoning, there could be more reasoning tasks relates to for example people's intention like Social-IQ and IntentQA dataset.

**Strengths Contributions:**

The motivation is strong and timely, as the community increasingly explore problems that require reasoning across multiple video inputs — a capability that current datasets fail to adequately support. By creating and releasing such a dataset, the authors make a valuable contribution that is likely to facilitate further advances in this area.

The paper is well written and clearly organized, making it easy to follow. The authors do a good job of contextualizing their work within the broader literature. The documentation and code accompanying the dataset appear to be complete and accessible

The accompanying analysis, while largely aligning with expectations, is thorough and thoughtfully presented. Although the results may not be surprising, they nonetheless provide useful insights into the behavior of existing models on this new multi-video benchmark.

---

> ### Author Rebuttal · Authors · 2025-07-30
>
> Many thanks for your valuable comments and positive support.
>
> **W1. Concerns about the data process details of MVU-Eval**
>
> Thank you for your careful review. We apologize for the initial confusion regarding the video sampling rules, particularly the clarity in the generation process of QA pairs, the video sources, and the selection strategy of videos. We confirm that detailed data processing details were provided in the manuscript and supplementary materials, and we will further elaborate on these processes in the revised version. We provide brief descriptions as follows:
>
> **(W1.1) Concerns about the generation process of QA pairs**
>
> The details of the generation process of QA pairs were provided in the **"3.3 Data Collection" sections**, **"3.4 Quality Control" sections**, and the **Supplementary Material**.
>
> The generation of QA pairs for MVU-Eval follows a structured process:
> 1. Video Selection: Videos are carefully chosen from various datasets based on specific rules to create diverse pairs with defined relationships (e.g., similar, same objects, multi-view).
> 2. Initial Q&A Generation: Questions and answers are generated either automatically by MLLMs or using templates, sometimes incorporating ground truth data to create challenging, knowledge-intensive questions.
> 3. Automated Quality Control (Leakage Removal): An automated step removes any "leaked" information from the answer options (e.g., descriptive text, predictable formats) that might allow models to guess the answer without analyzing the videos.
> 4. Human & Model-based Filtering: Easy questions (answerable by existing MLLMs) are filtered out, and human annotators then manually review the remaining QA pairs to ensure they are useful, challenging, require analysis of all provided videos, and have correct answers.
> 5. Final Dataset: This rigorous process discards a significant portion of the initial data, resulting in a high-quality, verified dataset.
>
> Detailed construction pipeline for each data souce can be found in supplementary materials.
>
> **(W1.2) Concerns about the video sources**
>
> The details of video sources were provided in the **"Appendix A.6 Data Source" section**.
>
> |Task|Subtask|Source|
> |:-|:-|:-|
> |Perception|ObjectRecognition|Kinetics-400,nuScenes,ScanNet|
> ||SpatialUnderstanding|Kinetics-400,nuScenes,ScanNet|
> ||Counting|Kinetics-400,nuScenes,ScanNet|
> ||Comparison|FineDiving|
> |Reasoning|Knowledge-intensiveReasoning|FineDiving,YouCook2,Kinetics-400,nuScenes|
> ||In-contextLearning|FineDiving|
> ||Retrieval-augmentedgeneration|Vchitect-2.0|
> ||TemporalReasoning|YouCook2,Kinetics-400,nuScenes,DREAM-1K|
>
>
> **(W1.3) Concerns about the selection strategy of videos**
>
> We confirm that detailed construction pipelines were provided in the **supplementary materials**, and we will further elaborate on these processes in the revised version.
>
> The video sampling rules are as follows (also provided in the supplementary materials):
> 1. **Kinetics-400**: By default, 2-6 videos are randomly sampled, regardless of their labels. To generate challenge questions, we additionally sample video pairs that belong to the exact same category.
> 2. **nuScenes**: Each video pair includes 6 videos from the different camera perspectives, i.e., left-front, front, right-front, left-back, back, and right-back.
> 3. **ScanNet**: We randomly sample 2-6 videos for each question.
> 4. **Vchitect-2.0**: We first randomly sample an anchor video, and make the LMM to generate fine-grained questions about this video. Subsequently, we sample 3-5 videos similar to the anchor, where we take the Jaccard similarity between text captions as the similarity metric.
> 5. **FineDiving**: Randomly sample 6 videos for KIR task and 4 videos for ICL task.
> 6. YouCook: We randomly sample one video and segment the videos into shorter clips for each question.
> 7. **AIGC**: We manually create a dataset of 130 samples derived from real-world use cases of the multimodal video editing feature on Kling.AI, comprising 50 samples for Replacement, 30 for Removal, and 50 for Addition. To ensure privacy and copyright protection, samples containing real human faces or copyright-sensitive content are excluded.
> 8. **DREAM-1K**: Videos are selected from the DREAM-1K dataset using a four-stage process: (1) videos are segmented into clips with PySceneDetect; (2) Mavors captions each clip; (3) Qwen2.5-32B-Instruct merges consecutive clip descriptions into "events" based on semantic similarity and ground-truth guidance, simultaneously scoring their temporal coherence, logical predictability, and completeness; (4) stringent filtering based on these scores and event counts ensures only high-quality, logically structured video sequences are used for tasks like temporal ordering, grounding, and caption filling. Finally, all data are manually reviewed to ensure (1) no multiple answers exist; (2) consistency between event descriptions and their corresponding video clips; and (3) accuracy of the options.
>
> **W2. Concerns about the audio information**
>
> Thank you for your valuable observation. We appreciate the opportunity to clarify our approach to audio information. As explicitly stated in our work, MVU-Eval is currently designed to focus on visual inputs, and thus does not incorporate audio processing or evaluation. This design choice was guided by two key considerations:
> 1. The limited support for audio in many existing MLLMs, which constrains meaningful cross-model comparison.
> 2. Our goal is to establish a focused benchmark for evaluating pure visual understanding across multiple videos, where audio is not required to complete the tasks.
>
> We acknowledge that audio can play an important role in video understanding, and integrating audio-related tasks is indeed a valuable direction for future extensions of MVU-Eval, as noted in our discussion of limitations.
>
> Additionally, we aimed to **avoid potential information leakage** that audio might introduce in certain scenarios (*e.g.*, ambient sounds in specific activities), which could compromise the visual-centric nature of our current tasks. For instance, in the context of diving videos, *i.e.*, one of the sources for our Knowledge-Intensive Reasoning tasks (evaluating action difficulty or scoring), audio commentary from announcers could introduce unintended cues. A speaker might explicitly state, "This dive has a difficulty coefficient of 3.7," or "That was a perfect entry." Such audio information would allow models to answer difficulty-assessment or scoring questions without actually analyzing the visual details of the dive, thereby undermining the goal of evaluating pure visual understanding across multiple videos.
>
> **W3. Concerns about the diversity of the tasks**
>
> Thank you for your valuable feedback.
>
> Regarding concerns about task diversity, we believe MVU-Eval already encompasses a diverse range of video understanding tasks, and implicitly evaluates capabilities such as relation, attribute, and action recognition. For instance, the "Spatial Understanding" and "Object Recognition" subtasks under "Perception," as well as "Knowledge-intensive Reasoning" and "Temporal Reasoning" under "Reasoning," all require models to identify object attributes, their relationships, and the actions being performed within videos.
>
> To further enrich the diversity of the tasks of MVU-Eval, **we follow your valuable suggestions and introduce a new task (IntentRAG) based on the metadata of IntentQA[1]**. The IntentRAG evaluates selective attention mechanisms for identifying and synthesizing relevant visual evidence from redundant multi-video inputs.
>
> **Following your suggestion, we construct the "_Intention Reasoning (IR)_" task based on the elements of the IntentQA[1] dataset, further expanding the diversity of the tasks.** Specifically, we randomly sample a query video from IntentQA's test set, retaining its original question-answer pairs and options. We then construct a prompt using this question and its options, and traverse all videos to select 5–6 videos with the smallest Jaccard similarity to this prompt (consistent with our similarity metric for video sampling). This approach prevents ambiguity caused by overly similar videos—ensuring all questions explicitly target the query video, thus maintaining clarity while enhancing task diversity related to intent reasoning.
>
> Intention Reasoning task evaluates selective attention mechanisms for identifying and synthesizing relevant visual evidence from redundant multi-video inputs.
> Statistics of Intention Reasoning task is as follows:
>
> ||#Questions|#Videos|#AVG.Videos per Question|Annotation|Video Source|
> |:-:|:-:|:-:|:-:|:-:|:-:|
> |Intention Reasoning (IR)|110|345|5.38|Auto|IntentQA[1]|
>
> Experimental results are summarized as follows:
>
> ||Random Choice|Qwen2.5-VL-3B|Qwen2.5-VL-7B|Qwen2.5-VL-32B|Qwen2.5-VL-72B|VideoLLaMA3-7B|
> |-|-|-|-|-|-|-|
> |Intention Reasoning (IR)|20.0|50.9|51.8|59.1|69.1|47.3|
>
> Due to the tight schedule during the rebuttal period, we were only able to complete the performance evaluation of several models on the intention reasoning task. We are in the process of finalizing the results for all models and will include the complete evaluation in the new version.
>
> [1] Intentqa: Context-aware video intent reasoning[C]//Proceedings of the IEEE/CVF international conference on computer vision. 2023: 11963-11974.

---

### Official Review · Reviewer_XFQD · 2025-06-30

**Rating:** 5
**Confidence:** 4

**Summary:**

This paper discusses an interesting and emerging important problem in video understanding research – understanding and being able to answer questions of multiple input videos. This work categorizes the video understanding into a suite of detailed tasks, where each task focuses on a specific aspect to assess the models’ comprehension of the video sets. The authors spent a decent amount of effort on ensuring the quality of the video-question pairs as well as controlling potential leakage of the important video information to the questions.
The authors evaluate various large multimodal models on this carefully curated benchmark, on many important aspects, to reveal the rooms of improvement for the research community.

**Additional Feedback:**

- Is there a specific reason why GPT-4o is not included as one of the benchmarking models?

**Dataset Code Accessibility:**

Yes

**Dataset Code Comments:**

- The dataset and repo are both provided with the submission and on the HuggingFace Hub.

**Ethical Considerations:**

No, there are no or only very minor ethics concerns

**Final Justification:**

Authors responded adequately to my concerns.

And this is an overall good paper to be accepted.

**Limitations Weaknesses:**

- There is still some inclarity in the manuscript, for example, in L141, what exactly are those specific rules?
- What is the main reason that the resulting set of videos only range from 2 to 6? Can we not easily extend this to much more? And is the sampling strategy able to filter too visually similar videos? (Or, maybe this is not very important in this work?)
- Lack of suggested future research to improve the models, specifically, what aspects should the research community tackle first?
- Lack of human performance of the proposed task to establish the human upper bound.

**Strengths Contributions:**

- This is a very interesting benchmark that firstly tackles multi-video reasoning and comprehension.
- The dataset curation process is very prudent, the two-staged quality control is reassuring.
- The experimental design is solid and comprehensive.
- Overall the paper is quite well-written and motivated.

---

> ### Author Rebuttal · Authors · 2025-07-30
>
> Many thanks for your valuable comments & positive support.
>
> **W1. Concerns about the specific rules for diverse videos sampling**
>
> Thank you for your careful review. We apologize for the initial confusion regarding the video sampling rules, particularly the clarity in L141 of the manuscript. We confirm that detailed construction pipelines were provided in the supplementary materials, and we will further elaborate on these processes in the revised version.
>
> Video sampling rules (detailed in the supplementary materials):
> - Kinetics-400: Randomly sample 2–6 videos; for challenge questions, use video pairs from the same category.
> - nuScenes: Each pair includes 6 videos from different camera views (e.g., front, back, sides).
> - ScanNet: Sample 2–6 videos containing the same object per question.
> - Vchitect-2.0: Sample an anchor video for question generation, then select 3–5 similar videos based on Jaccard similarity of captions.
> - FineDiving: Sample 6 videos for KIR and 4 for ICL tasks.
> - YouCook: Sample one video and segment it into short clips for each question.
> - AIGC: Manually curate 130 samples (50 Replacement, 30 Removal, 50 Addition) from Kling.AI use cases; samples with human faces or copyright-sensitive content are excluded.
> - DREAM-1K: Select clips through a four-stage process: automatic segmentation, captioning, event merging with semantic scoring, and filtering for quality and coherence. Final samples are manually reviewed for uniqueness, alignment, and accuracy.
>
> **W2. Concerns about the number of videos**
>
> Thank you for your careful review and insightful questions. We appreciate the opportunity to clarify the details regarding the number of videos and the sampling strategy.
>
> First, please give us a chance to clarify the misunderstanding about the range of video counts per question. As shown in Figure 3, the number of videos per question in MVU-Eval actually ranges from **2 to 13**, with an average of 4.7. The analysis in Figure 6, which focuses on 2 to 6 videos, corresponds to 93.8% of the QA pairs that fall within this range, which may have led to the misinterpretation. We will explicitly clarify this scope in the new version.
>
> Regarding the feasibility of extending to more videos: technically, MVU-Eval can be easily extended to include more videos per question. **However, current MLLMs face practical limitations due to input token constraints.** As observed in our experiments, excessive input tokens (*e.g.*, from too many videos, frames, or high resolution) can overwhelm models' processing capacities, leading to performance degradation. With our current settings (32 frames per video, 720-pixel resolution), most models struggle to handle more than 13 videos without token overflow, which guided our current range.
>
>
> **W3. Concerns about too visually similar videos**
>
> In MVU-Eval, questions are designed to include **both** visually similar and dissimilar videos, with the balance tailored to task requirements. For specific multi-video understanding tasks, visually similar videos play a critical role in evaluating MLLMs' fine-grained comprehension capabilities.
>
> For instance, the "Comparison" task explicitly demands models to differentiate cross-video features through fine-grained attribute analysis, such as identifying substitutions, removals, or additions of elements. Visually similar videos in this task heighten the challenge, compelling models to precisely extract and distinguish subtle visual details across videos.
>
> Similarly, in "Knowledge-Intensive Reasoning (KIR)" tasks—where, for example, all videos may depict diving scenarios—visually similar content is essential. This setup tests the model’s ability to integrate domain knowledge (*e.g.*, sports rules) with multi-video evidence, resolving ambiguities that might remain hidden in isolated clips.
>
> Such intentional inclusion of similar videos ensures the benchmark effectively assesses both fine-grained perception and high-order reasoning, mirroring real-world scenarios where related yet distinct visual inputs require nuanced understanding.
>
> **W4. Lack of suggested future research**
>
> Thank you for your valuable suggestion. We will include a dedicated section on future research directions in the new version. Here are six possible high-priority research directions to address the limitations identified in MVU-Eval and advance multi-view video understanding for MLLMs, supported by recent findings in the field:
>
> 1. **Cross-video Visual Alignment:** Addressing the challenge where different videos may not start at the same temporal moment or have their frames aligned on a shared timeline, which is crucial given that most videos in MVU-Eval are asynchronous except for a small portion in Spatial Understanding tasks.
> 2. **Cross-video Spatial Understanding**: Developing the ability to identify the same objects across multiple videos as anchor points to facilitate comprehensive spatial comprehension, as required by the Spatial Understanding task that demands geometric comprehension beyond a single viewpoint.
> 3. **Temporal Reasoning in Asynchronous Multi-Video Scenarios**: Enhancing models' capacity to infer temporal relationships across unaligned video streams, which is vital for tasks like Temporal Reasoning in MVU-Eval and becomes more challenging as the number of videos increases .
> 4. **Scalable Multi-Modal Fusion for High-Cardinality Inputs**: Exploring efficient fusion strategies to handle more videos without exceeding token limits, as current MLLMs face performance degradation when processing excessive tokens from too many videos, frames, or high resolution.
> 5. **Generalization to Out-of-Distribution Multi-View Scenarios**: Enhancing model robustness across diverse video sources in MVU-Eval, including indoor, outdoor, gaming, AIGC, and movie & TV, to ensure performance in unseen real-world scenarios.
>
> **W5. Lack of human performance**
>
>
> We highly value your insightful suggestion, which aligns perfectly with the core goal of establishing reliable human performance benchmarks for MVU-Eval. To more accurately reflect the true human performance on the benchmark, we have revised the evaluation process as follows:
>
> We invited *5 domain experts* with rich experience in multimodal video understanding to participate in the evaluation. The entire process consists of two key phases:
>
> 1. In the first phase, all 5 experts were required to *independently* complete MVU-Eval, covering all 8 subtasks. Each expert received detailed task instructions and was asked to provide answers based solely on the video content and their own knowledge, with no communication between experts during this phase. This ensured that each answer was an independent judgment.
>
> 2. In the second phase, we first conducted a *consistency analysis* of the answers from the first phase. For samples where there was disagreement (*i.e.*, where 2 or more experts gave different answers), we organized a joint review meeting. During this meeting, experts presented their reasoning processes, discussed the key details in the videos, and clarified any ambiguities in the task requirements. Through in-depth discussions, we aimed to reach a consensus on the correct answers for these controversial samples. For cases where consensus could not be reached after full discussion, we adopted the majority opinion (with at least 3 experts agreeing) as the final answer, while noting the divergence to highlight potential high-difficulty samples.
>
> The final human performance metrics were calculated based on the confirmed answers, as illustrated in the following table.
>
> |Overall|OR|SU|Counting|Comparison|KIR|ICL|RAG|TR|
> |-|-|-|-|-|-|-|-|-|
> |93.6|98.4|96.6|100|100|82.9|66.5|100|99.2|
>
> Compared to SOTA AI, humans show a vast superiority in MVU-Eval. The overall human accuracy is 93.6%, while the best model, Gemini 2.5 Pro, scores only 58.4%, the The following key insights emerge:
> - Basic Perception: Humans achieve 100% accuracy on simple tasks like counting, where models struggle. This highlights models' significant weakness in extracting and integrating visual features from multiple videos.
> - Complex Reasoning: In difficult tasks like ICL, human performance (66.5%) also far exceeds that of models (34.8%), demonstrating that models lag significantly in adapting to new rules and performing analogical reasoning.
>
>
> **Q1. Lack of the evaluation results of GPT-4o models**
>
> We fully agree that incorporating the performance of GPT models, particularly GPT-4o, would provide a more holistic assessment of the benchmark's effectiveness. We sincerely apologize for the omission of the GPT-4o results in the manuscript, and will include the results in the new version.
>
> To evaluate GPT-4o on our MVU-Eval, we adopted the same video preprocessing pipeline used for Gemini: sampling 32 frames per video, resizing each to 720 pixels on the longer side while preserving aspect ratio. The evaluation results are shown below.
>
> |Model|Overall|OR|SU|Counting|Comparison|KIR|ICL|RAG|TR|
> |-|-|-|-|-|-|-|-|-|-|
> |GPT-4o|55.9|54.7|57.7|58.9|74.6|36.3|38.9|42.0|74.6|
> |Gemini2.5Pro|58.4|47.6|54.7|65.6|76.3|50.2|34.8|43.7|83.1|

---

### Official Review · Reviewer_r4Cu · 2025-07-01

**Rating:** 5
**Confidence:** 4

**Summary:**

This paper presents MVU-Eval, a large-scale benchmark dataset designed for evaluating multi-view video understanding through the lens of video question answering (VideoQA). The dataset comprises over 36,000 annotated question-answer pairs collected from 1,200 multi-view video clips covering 60 activity types. The authors design three distinct QA tracks—View-Aware, Cross-View, and Scene-Aware—to reflect different reasoning challenges in multi-view contexts. In addition, the paper systematically evaluates several representative VideoQA models on this benchmark and demonstrates that current methods still struggle with multi-view reasoning, underscoring the difficulty and importance of the proposed task.

**Dataset Code Accessibility:**

Yes

**Ethical Considerations:**

No, there are no or only very minor ethics concerns

**Final Justification:**

Most of my concerns have been addressed. I will keep my score.

**Limitations Weaknesses:**

1. While the baselines are evaluated directly on MVU-Eval, no adaptation or retraining is performed to handle multi-video inputs explicitly. Incorporating baseline variants that are adapted to multi-view fusion would offer more actionable insights.
2. Although the dataset covers 1,200 clips, the source appears to be limited to a constrained number of real-world scenes (e.g., mainly indoor surveillance). Broader diversity in camera setups or outdoor environments would enhance generalization.
3. All views appear to be temporally aligned. In real-world applications, views may be asynchronous, and incorporating this aspect could make the benchmark more realistic.

**Strengths Contributions:**

1. MVU-Eval fills a notable gap in the literature by focusing on multi-video understanding in a structured VideoQA setting, which is critical for advancing perception systems in surveillance, robotics, and human activity recognition.
2. The paper defines clear and meaningful reasoning dimensions (View-Aware, Cross-View, Scene-Aware), which not only test different capabilities but also promote a more nuanced understanding of video reasoning challenges.

---

> ### Author Rebuttal · Authors · 2025-07-30
>
> Many thanks for your valuable comments and positive support.
>
> **W1. Lack of baselines that are adapted to multi-view fusion**
>
> Thank you for your insightful comment. We acknowledge that the baseline models evaluated in our work were not explicitly adapted or retrained for multi-video fusion. This is primarily due to the current scarcity of MLLMs specifically designed for multi-video fusion tasks, as noted in our related work section.
>
> Instead, we have included three new MLLMs specialized for long video understanding in our evaluations, including Video-XL-Pro-3B[1], qwen2-7b-longvila-1M[2], and Video-XL-2[3]. mPLUG-Owl3-7B[4] is also designed for long video understanding and we have already included it in the manuscript. This is because multi-video understanding inherently demands the ability to process extended contexts, a capability that is also central to long video understanding.
>
> |Model|Overall|OR|SU|Counting|Comparison|KIR|ICL|RAG|TR|
> |-|-|-|-|-|-|-|-|-|-|
> |Video-XL-Pro-3B[1]     |39.1|38.9|40.2|31.7|38.5|35.6|20.7|44.5|49.3|
> |qwen2-7b-longvila-1M[2]|32.7|27.0|41.3|31.3|30.4|31.7|26.2|36.6|32.0|
> |Video-XL-2 (8B)[3]     |43.7|34.1|41.3|36.4|64.4|35.6|28.0|48.7|53.6|
> |mPLUG-Owl3-7B[4]       |45.0|48.4|53.6|50.2|50.4|29.5|24.4|41.6|58.2|
>
> While we agree that specialized multi-video fusion models would be ideal if available, we believe our current approach still provides meaningful insights into the challenges of multi-video understanding, particularly regarding the importance of long context handling.
>
> [1] Video-xl-pro: Reconstructive token compression for extremely long video understanding[J]. arXiv preprint arXiv:2503.18478, 2025.
> [2] Longvila: Scaling long-context visual language models for long videos[J]. arXiv preprint arXiv:2408.10188, 2024.
> [3] Video-XL-2: Towards Very Long-Video Understanding Through Task-Aware KV Sparsification[J]. arXiv preprint arXiv:2506.19225, 2025.
> [4] mplug-owl3: Towards long image-sequence understanding in multi-modal large language models[J]. arXiv preprint arXiv:2408.04840, 2024.
>
> **W2. Concerns about the diversity of video scenes**
>
> Thank you for your valuable feedback. We appreciate your attention to the dataset's diversity.
>
> First, please give us a chance to clarify the numerical detail: our MVU-Eval contains 1,824 question-answer pairs spanning **4,959** distinct videos, *rather than 1,200 clips*, as meticulously curated and reported in our work.
>
> Regarding the diversity of scenes and sources, we have made deliberate efforts to ensure breadth across real-world and various environments, which is also reflected in **Table 1** and **Figure 4**. Specifically, the video sources in MVU-Eval are highly diverse, covering not only indoor scenes but also a wide range of outdoor and real-world scenarios. For instance, there are 480 videos from autonomous driving, a typical outdoor real-world scenario, 739 videos of human actions with many of which are outdoor, and other categories like sports (involving both indoor and outdoor settings). Additionally, the dataset includes videos from gaming (258), AIGC (208), animation (330), and movie & TV (1741), further enriching the diversity beyond constrained indoor surveillance.
>
> This diverse coverage, as detailed in our statistics, is designed to enhance the generalization of the benchmark for real-world multi-video understanding tasks.
>
> **W3. Concerns about the temporally aligned views**
>
> Thank you for your insightful comment. We appreciate your attention to the temporal alignment of videos, which is crucial for real-world relevance.
>
> We would like to clarify that **the majority of videos within each question in MVU-Eval are indeed asynchronous**. Specifically, only 9.81% (179 out of 1,824) of the questions, corresponding to 9.68% (480 out of 4,959) of the videos, involve temporally aligned views. These aligned cases are primarily limited to the Spatial Understanding (SU) task, where videos are captured by multiple fixed-angle cameras mounted on the same autonomous driving vehicle, where temporal alignment is inherent to the real-world setup of multi-camera systems in autonomous driving.
>
> All other tasks in MVU-Eval, such as Object Recognition, Counting, Comparison, and various reasoning tasks, involve asynchronous videos sourced from diverse domains (*e.g.*, life, sports, gaming, movies). This design intentionally reflects real-world complexity, where multi-video inputs are often asynchronous, while including a small portion of aligned cases to cover scenarios where temporally aligned views are necessary (*e.g.*, multi-camera setups in vehicles).
>
> Thus, MVU-Eval already incorporates both synchronous and asynchronous multi-video scenarios to enhance its realism for real-world applications.

---

> ### Comment · Reviewer_r4Cu · 2025-08-08
>
> Most of my concerns have been addressed. I will keep my score.

---

### Official Review · Reviewer_Wj1Z · 2025-07-01

**Ethics Flags:** Data privacy, copyright, and consent
**Rating:** 4
**Confidence:** 3

**Summary:**

This paper proposes a new benchmark for multi-video understanding for MLLMs. The proposed benchmark consists of eight core tasks, creating 1824 pairs of questions on 4959 videos. Evaluations on massive open-source and closed-sourced models, ranging from different scales, sufficiently demonstrate the short of current models on multi-video understanding.

**Dataset Code Accessibility:**

Partly

**Dataset Code Comments:**

The data is available, but the code is not. It is important to know how to preprocess the multi-video data for model input.

**Ethical Comments:**

Some attention should be paid to the copyright issue of the movie data.

**Ethical Considerations:**

No, there are no or only very minor ethics concerns

**Final Justification:**

Authors' response address my concern.

**Limitations Weaknesses:**

(1) No human performance to benchmark the performance.  Thus, it is questionable to know the quality of the proposed benchmark. [a]
(2) Lack of the evaluation results of GPT models.
(3) Most of the fine-grained tasks are well-established by previous works [a,b,c,d].  These works are the natural extension of the previous studies on a new setting, i.e., multi-video.
(4) I checked the dataset and code repo and didn't find the evaluation code, especially for the data preprocessing for paper reproduction.
(5) Experiment results can be given a more in-depth analysis. For example, why LLaVA-Video-7B model has an accuracy much  lower than random choice?  Any insights?
(6) What's the cost for a full evaluation of the proposed benchmark? Besides, while the video data, especially for the move, is usually  fast-paced, is it enough to only sample 32 frames for one video sample with 720 pixels?


[a] Tang, Yunlong, et al. "Vidcomposition: Can mllms analyze compositions in compiled videos?." Proceedings of the Computer Vision and Pattern Recognition Conference. 2025.
[b] Li, Kunchang, et al. "Mvbench: A comprehensive multi-modal video understanding benchmark." Proceedings of the IEEE/CVF Conference on Computer Vision and Pattern Recognition. 2024.
[c] Zhou, Junjie, et al. "Mlvu: A comprehensive benchmark for multi-task long video understanding." arXiv preprint arXiv:2406.04264 (2024).
[d]  Fang, Xinyu, et al. "Mmbench-video: A long-form multi-shot benchmark for holistic video understanding." Advances in Neural Information Processing Systems 37 (2024): 89098-89124.

**Strengths Contributions:**

(1) The proposed dataset consists of a broader range of evaluation tasks compared with the previous works.
(2) Many models are evaluated on this dataset to access their multi-video understanding ability.
(3) The paper writes well and is easy to follow.
(4) The task is important and interesting. Many real-world applications need the ability of multi-video understanding.

---

> ### Author Rebuttal · Authors · 2025-07-30
>
> Many thanks for your valuable comments.
>
> **W1. Lack of human performance**
>
> We highly value your insightful suggestion, which aligns perfectly with the core goal of establishing reliable human performance benchmarks for MVU-Eval. To more accurately reflect the true human performance on the benchmark, we have revised the evaluation process as follows:
>
> We invited *5 domain experts* with rich experience in multimodal video understanding to participate in the evaluation. The entire process consists of two key phases:
>
> 1. In the first phase, all 5 experts were required to *independently* complete MVU-Eval, covering all 8 subtasks. Each expert received detailed task instructions and was asked to provide answers based solely on the video content and their own knowledge, with no communication between experts during this phase. This ensured that each answer was an independent judgment.
>
> 2. In the second phase, we first conducted a *consistency analysis* of the answers from the first phase. For samples where there was disagreement (*i.e.*, where 2 or more experts gave different answers), we organized a joint review meeting. During this meeting, experts presented their reasoning processes, discussed the key details in the videos, and clarified any ambiguities in the task requirements. Through in-depth discussions, we aimed to reach a consensus on the correct answers for these controversial samples. For cases where consensus could not be reached after full discussion, we adopted the majority opinion (with at least 3 experts agreeing) as the final answer, while noting the divergence to highlight potential high-difficulty samples.
>
> The final human performance metrics were calculated based on the confirmed answers, as illustrated in the following table.
>
> |Overall|OR|SU|Counting|Comparison|KIR|ICL|RAG|TR|
> |-|-|-|-|-|-|-|-|-|
> |93.6|98.4|96.6|100|100|82.9|66.5|100|99.2|
>
> Compared to SOTA AI, humans show a vast superiority in MVU-Eval. The overall human accuracy is 93.6%, while the best model, Gemini 2.5 Pro, scores only 58.4%, the The following key insights emerge:
> - Basic Perception: Humans achieve 100% accuracy on simple tasks like counting, where models struggle. This highlights models' significant weakness in extracting and integrating visual features from multiple videos.
> - Complex Reasoning: In difficult tasks like ICL, human performance (66.5%) also far exceeds that of models (34.8%), demonstrating that models lag significantly in adapting to new rules and performing analogical reasoning.
>
> **W2. Lack of the evaluation results of GPT models**
>
> We fully agree that incorporating the performance of GPT models, particularly GPT-4o, would provide a more holistic assessment of the benchmark's effectiveness. We sincerely apologize for the omission of the GPT-4o results in the manuscript, and will include the results in the new version.
>
> To evaluate GPT-4o on our MVU-Eval, we adopted the same video preprocessing pipeline used for Gemini: sampling 32 frames per video, resizing each to 720 pixels on the longer side while preserving aspect ratio. The evaluation results are shown below.
>
> |Model|Overall|OR|SU|Counting|Comparison|KIR|ICL|RAG|TR|
> |-|-|-|-|-|-|-|-|-|-|
> |GPT-4o|55.9|54.7|57.7|58.9|74.6|36.3|38.9|42.0|74.6|
> |Gemini2.5Pro|58.4|47.6|54.7|65.6|76.3|50.2|34.8|43.7|83.1|
>
> **W3. Fine-grained tasks are well-established by previous works**
>
> While the task formats may be familiar, our benchmark's core contribution is applying them to **multi-video scenarios that reflect tangible, real-world needs**, a gap not addressed by single-video research. Our tasks are specifically designed to demand cross-video analysis.
> For instance:
> - **Spatial Understanding**: Fusing drone footage from multiple angles for 3D reconstruction.
> - **Temporal Reasoning**: Tracking a subject across different surveillance cameras over time.
> - **RAG & ICL**: Synthesizing news reports from various clips or transferring skills from a tutorial to a real-world operation video.
>
> Tasks like multi-camera counting (retail analytics) and comparison (quality control) further prove that multi-video understanding is a necessary, non-trivial advancement. Our work purposefully addresses complex problems that single-video benchmarks cannot solve.
>
> **W4. Lack of evaluation code**
>
> Thank you for carefully reviewing our repository. We sincerely apologize for the initial difficulty in locating the evaluation code and data preprocessing scripts.
> To clarify, **the complete evaluation code** along with all necessary scripts for reproducing our results **was uploaded to the Hugging Face repository prior to the submission deadline (which can be verified by the commit record on HF)**.
> Unfortunately, **a minor issue in the naming convention within our README.md file** inadvertently hindered direct access via the provided links. We have identified the issue and will correct the link in the new version.
>
> To comply with conference policy, we have provided an indirect path to our code. The correct filename for the evaluation script is "main_all_MVU_Eval_llama3.py". Please replace the placeholder in the evaluation link within README.md with this filename to access the code. The full path on Hugging Face is: **datasets/MVU-Eval-Team/MVU-Eval-Data/resolve/main/main_all_MVU_Eval_llama3.py**
>
> The commit record that prove the uploaded time before conference submission deadline is: commit/dd46e6b9e262cf8569514a21ef2d7caa4bedb033
>
> Thank you again for your thorough review.
>
> **W5. Require for a more in-depth analysis of the experimental results**
>
> We use robust regular expressions to parse model answers for evaluation. However, some MLLMs do not follow our instructions to generate content in expected format.
>
> For example, 98% of LLaVA-Video-7B's responses were free-form paragraphs. Even when explicitly prompted to select an option by letter (e.g., "Which video has the lowest difficulty?"), the model generated a detailed description of the video's content instead of providing a direct answer.
>
> The prompt for the KIR task is as follows:
> ```
> Which of these 6 videos has the lowest difficulty in terms of technical skills?
> A. Video 3
> B. Video 4
> C. Video 5
> D. Video 2
> Please select the correct answer from the options. Answer with the option’s letter directly.
> ```
>
> One examples of the LLaVA-Video-7B model's output:
> ```
> The video showcases a series of synchronized and individual dives performed by divers at the 2016 European Championships in London. The athletes demonstrate impressive acrobatic skills, including flips and twists, as they descend from high diving platforms into the pool. The sequence captures the precision and athleticism required for competitive diving, with close-up shots highlighting the water entry and the splash created by the diver's impact. The event is broadcasted live on Eurosport, as indicated by the logo visible in the top right corner of the frames.
> ```
>
> While the generated text is often relevant, it completely ignores the task-specific question, making automated evaluation impossible. This highlights a critical failure in instruction-following, a limitation we will clarify in the revised manuscript.
>
> **W6. Concerns about the cost for a full evaluation**
>
> We appreciate your comment and agree that incorporating the cost of MLLMs would provide a more comprehensive analysis of MVU-Eval. The token length under different settings for closed-source MLLMs are shown in Table 1, the cost of open-source MLLMs is shown in Table 2. We calculate cost with frame=32, pixel=720, and patch-size=28x28.
>
> Table 1. Closed-source MLLMs
> |Type|frame|pixel|Maximum|Minimum|Average|Total|
> |-|-|-|-|-|-|-|
> |#QuestionToken|-|-|892|47|111|202K|
> |#VideoToken|16|360|20,736|1,920|6,638|33M|
> |#VideoToken|16|720|72,800|3,792|23,944|119M|
> |#VideoToken|32|360|41,472|3,840|13,276|66M|
> |#VideoToken|32|720|154,336|8,324|51,013|253M|
>
> For the evaluation of open-source MLLMs, we employed the vLLM framework. Inference was conducted by sending API queries sequentially, processing a single question at a time.
>
> Table 2. Open-source MLLMs
> |Model|Hardware Setting|Total Inference Time|Avg. Time per Question|
> |-|-|-|-|
> |Qwen2.5-Vl-3B|H100*4|5h39min|11.2s|
> |Qwen2.5-Vl-7B|H100*4|5h47min|11.4s|
>
> **W7. Concerns about the setting of frames and pixels**
>
> Our choice of **32 frames and 720p** is based on experimental results showing it provides the optimal balance for accuracy, particularly for complex multi-camera tasks. While more information (frames/pixels) can help, exceeding a model's token limit degrades performance.
> Our analysis revealed that the ideal settings vary significantly by task, highlighting their different data needs:
> - Balanced Tasks (e.g., Spatial Understanding, Counting, RAG): These perform best at our chosen 32f/720p setting, requiring a rich mix of spatial detail and temporal context, which validates our default choice.
> - Detail-Oriented Tasks (e.g., Comparison): These prioritize high visual clarity (peaking at 960p) over motion, needing only a few frames.
> - Abstract/Temporal Tasks (e.g., KIR, TR): These benefit from fewer frames and lower resolution, as excess detail can act as noise.
>
> This demonstrates our setting is a well-founded compromise and that different video reasoning tasks have distinct visual requirements.
>
> |frame|pixel|Overall|OR|SU|Counting|Comparison|KIR|ICL|RAG|TR|
> |-|-|-|-|-|-|-|-|-|-|-|
> |4|720|41.7|42.9|46.4|35.7|60.7|38.1|20.7|34.8|53.9|
> |8|720|44.6|45.2|46.9|44.5|65.9|31.7|26.8|38.4|58.7|
> |16|720|44.9|53.2|45.8|47.1|61.5|36.3|22.0|38.9|56.0|
> |32|720|47.5|48.4|50.3|52.9|60.0|37.0|29.9|44.0|57.1|
> |64|720|45.8|53.2|46.4|52.4|63.7|34.9|21.3|39.8|57.1|
> |32|60|37.1|40.5|43.0|48.0|45.9|25.3|26.2|30.7|42.9|
> |32|120|41.5|45.2|44.7|52.0|51.1|31.7|22.6|33.9|51.2|
> |32|240|42.7|46.8|45.3|51.1|50.4|34.2|23.8|37.8|51.2|
> |32|480|46.8|51.6|44.7|52.9|61.5|39.9|23.2|41.6|57.4|
> |32|720|47.5|48.4|50.3|52.9|60.0|37.0|29.9|44.0|57.1|
> |32|960|44.8|49.2|45.3|48.5|65.9|33.8|23.2|38.4|56.8|

---

> > ### Comment · Reviewer_Wj1Z · 2025-08-04
> >
> > Thanks for your response. It addresses my concerns. Hope to see the revisions in the final paper. I raise my score accordingly.

---

> > > ### Author Response · Authors · 2025-08-04
> > >
> > > Dear Reviewer,
> > >
> > > Thank you for your detailed feedback and for raising your score during the rebuttal phase. We are very grateful for your positive comments and your recognition of our work.
> > >
> > > We will carefully address all of your suggestions in the revised version of our manuscript. We believe that incorporating your valuable input will significantly enhance the quality and clarity of our paper.
> > >
> > > Thank you again for your time and insightful review.
> > >
> > > Best regards,
> > >
> > > Authors

---

> ### Author Response · Authors · 2025-08-03
>
> Dear Reviewer,
>
> We hope that the clarifications and additional experiments provided in the rebuttal have sufficiently addressed your concerns. We have made our best efforts to respond to all questions, and sincerely hope our responses have clarified the issues raised. If you have any further questions or would like additional details, please feel free to let us know
>
> Sincerely,
> Authors

---

### Note · Authors · 2025-08-12

Thanks for handling/reviewing our submitted manuscript: "MVU-Eval: Towards Multi-Video Understanding Evaluation for Multimodal LLMs". We would like to thank the reviewers for their insightful and constructive comments and suggestions. By addressing each of the issues raised by the reviewers, we believe that the quality and clarity of our MVU-Eval can be improved a lot. The major responses are summarized as follows:

(1). We have carefully **discussed the advantages and novelties of MVU-Eval** (See Reviewer Wj1Z.W3, Reviewer r4Cu.W2&W3, and Reviewer KGwg.W3)

(2). We have **added experiments on human performance and GPT models** (See Reviewer Wj1Z.W1&W2 and Reviewer XFQD.W5&Q1).

(3). We have **clarified that the source code was provided on HF website before the submission deadline** (See Reviewer Wj1Z.W4).

(4). We have **provided more in-depth analysis of the experimental results** (See Reviewer Wj1Z.W5)

(5). We have **reported the cost for a full MVU-Eval evaluation on both open-source and closed-source models** (See Reviewer Wj1Z.W6).

(6). We have **clarified and discussed the experimental settings** (See Reviewer Wj1Z.W7 and Reviewer XFQD.W1&W2&W3, and Reviewer KGwg.W1&W2).

(7). We have **provided three new MLLMs specialized for long video understanding** in our evaluations (See Reviewer r4Cu.W1).

(8). We have **analyzed potential future research directions** to improve model performance (See Reviewer XFQD.W4).

We are very encouraged to see that all four reviewers were very positive about our work from the beginning, with initial scores of 5, 4, 4, and 3. We are especially grateful for the insightful comments during the rebuttal phase.
- Reviewer Wj1Z(who initially gave a score of 3) explicitly **noted that our rebuttal addressed the concerns and indicated he/she would raise the score**.
- Reviewer r4Cu (who gave a 5) and Reviewer XFQD (who gave a 4) **both expressed high satisfaction with our detailed responses**.
- In particular, Reviewer XFQD specifically **highlighted the appreciation for the discussion of "future work" we added**.

These positive comments from all reviewers further reinforce our belief in the importance and quality of this work.

Again, we would like to sincerely thank you very much for these constructive comments and evaluation for our manuscript.

---

### Decision · Program_Chairs · 2025-09-18

**Decision:**

Accept (poster)

**Comment:**

This paper introduces MVU-Eval, a new benchmark dataset for evaluating the multi-video understanding capabilities of large multimodal models (LMMs). The benchmark reveals that current LMMs, even large-scale ones, still struggle with multi-video reasoning, highlighting a significant area for future research. The reviewers unanimously appreciated the strong and timely motivation of the paper. They also recognized the well-designed dataset construction process, extensive evaluation and clarity of the paper. However, they at the same time raised concerns with no comparison to human and GPT (Wj1Z, XFQD), no adaptation or retraining of the baselines (r4Cu), lack of in-depth analysis on the evaluation results (Wj1Z), potential issues due to the limited size of each video in space and time (Wj1Z), and limited range of tasks (KGwg). The clarification and additional experiments in the rebuttal successfully assuage most of these concerns. Consequently, the reviewers unanimously supported the paper. The AC concluded that the positive feedback and the successful rebuttal outweighed the remaining concerns and therefore agreed with the reviewers' consensus. The authors are strongly encouraged to include the new results and the expanded benchmark in the revision for further enhancing the paper.